# Cell cycle-dependent gene networks for cell proliferation activated by nuclear CK2α complexes

Miwako Kato Homma[1], Ryuichiro Nakato[2], Atsushi Niida[3], Masashige Bando[4], Katsunori Fujiki[4], Naoko Yokota[2], So Yamamoto[1], Takeshi Shibata[5], Motoki Takagi[6], Junko Yamaki[1], Hiroko Kozuka-Hata[7], Masaaki Oyama[7], Katsuhiko Shirahige[4,8,9], Yoshimi Homma[1]

Nuclear expression of protein kinase CK2α is reportedly elevated in human carcinomas, but mechanisms underlying its variable localization in cells are poorly understood. This study demonstrates a functional connection between nuclear CK2 and gene expression in relation to cell proliferation. Growth stimulation of quiescent human normal fibroblasts and phospho-proteomic analysis identified a pool of CK2α that is highly phosphorylated at serine 7. Phosphorylated CK2α translocates into the nucleus, and this phosphorylation appears essential for nuclear localization and catalytic activity. Protein signatures associated with nuclear CK2 complexes reveal enrichment of apparently unique transcription factors and chromatin remodelers during progression through the $G_1$ phase of the cell cycle. Chromatin immunoprecipitation-sequencing profiling demonstrated recruitment of CK2α to active gene loci, more abundantly in late $G_1$ phase than in early $G_1$, notably at transcriptional start sites of core histone genes, growth stimulus-associated genes, and ribosomal RNAs. Our findings reveal that nuclear CK2α complexes may be essential to facilitate progression of the cell cycle, by activating histone genes and triggering ribosomal biogenesis, specified in association with nuclear and nucleolar transcriptional regulators.

## Introduction

Protein kinase CK2 is a ubiquitous eukaryotic Ser/Thr kinase that is indispensable for cell viability and that is involved in a vast array of physiological processes (Lozeman et al, 1990; Litchfield, 2003). It is composed of two catalytic subunits, α and α′ and regulatory β subunits that form a variable, tetrameric holoenzyme with the composition $α_2β_2$, $αα′β_2$, or $α′_2β_2$, in which depletion of the CK2α catalytic gene arrests yeast cells in $G_1/S$ and $G_2/M$ (Hanna et al, 1995). CK2 phosphorylates more than 300 substrates in eukaryotes (Meggio & Pinna, 2003), many of which are involved in cell proliferation, anti-apoptosis signaling, control of cell division (Rabalski et al, 2016), and development of regulatory T cells (Ulges et al, 2015; Gibson et al, 2018). CK2 had been thought to maintain high basal activity independent of second messengers; however, we demonstrated cell-cycle-dependent activation of CK2 and identified a direct target for CK2, a eukaryotic translational initiation factor, eIF5, through which CK2 may regulate protein synthesis in the early $G_1$ phase of normal fibroblasts (Homma et al, 2005). Concerning transcriptional elongation in a yeast system, phosphorylation of histone H2A by CK2 suggested a conserved role of CK2 in binding to gene bodies and enhancer regions throughout the genome (Basnet et al, 2014). A phosphoproteomic study, using a variety of protein kinase substrates specific for CK2, revealed a fundamental role of CK2 in the splicing machinery of spliceosomes (Bian et al, 2013). A phosphoproteomic study that exploited CRISPR/Cas9 technology generated viable clones devoid of either of the two CK2 catalytic subunits or β subunits in myoblasts. In these clones, the highly pleiotropic functions of CK2 greatly altered phospho-proteomic profiles (Borgo et al, 2017). Another study reported that mini-chromosome maintenance protein complex genes were decreased by down-regulation of CK2α using siRNA (Schaefer et al, 2019).

In relation to carcinogenesis, exogenous CK2α caused 9 of 37 transgenic mice to develop T lymphoma (Seldin & Leder, 1995). Most interestingly, CK2α mRNA and/or protein were invariably elevated in rapidly proliferating tissues and tumors, such as head and neck carcinomas and other solid tumors (Gapany et al, 1995; Landesman-Bollag et al, 2001; Sousa et al, 2023). In cells of such tissues, the ratio of nuclear versus cytosolic CK2α subunit was much higher than in a normal tissue (Russel et al, 1999). Still, other studies suggested that

[1]Department of Biomolecular Sciences, Fukushima Medical University School of Medicine, Fukushima, Japan  [2]Laboratory of Computational Genomics, Institute for Quantitative Biosciences, University of Tokyo, Bunkyo, Japan  [3]Human Genome Center, The Institute of Medical Science, The University of Tokyo, Minato, Japan  [4]Institute for Quantitative Biosciences, The University of Tokyo, Bunkyo, Japan  [5]K.K. ABSciex, Shinagawa, Japan  [6]Translational Research Center, Fukushima Medical University School of Medicine, Fukushima, Japan  [7]Medical Proteomics Laboratory, The Institute of Medical Science, The University of Tokyo, Minato, Japan  [8]Department of Biosciences and Nutrition, Karolinska Institutet, Biomedicum, Stockholm, Sweden  [9]Department of Cell and Molecular Biology, Karolinska Institutet, Biomedicum, Stockholm, Sweden

Correspondence: mkhomma@fmu.ac.jp

variable CK2 localization in cells is associated with a wide spectrum of malignancies (Chua et al, 2017). We recently demonstrated that enhanced nuclear or nucleolar localization of CK2α in breast cancer primary specimens is closely linked to poor prognosis (Homma et al., 2021, 2022). In order to understand molecular mechanisms and whether CK2 might be involved in downstream gene transcription, it is also important to address how its localization in cells is regulated and how kinase activities are controlled in those locations during progression of the cell cycle.

In this study, we investigated the molecular mechanism of CK2 activation in the nucleus and its role in gene transcription essential for progression of the cell cycle. We demonstrate that multiple phosphorylation sites on CK2α are associated with its nuclear translocation in human normal fibroblast, leading to gene activation by functional nuclear CK2 complexes. Specifically, chromatin immunoprecipitation-sequencing (ChIP-seq) analysis, using CK2α-CRISPR/Cas9 knockout cells as a control, revealed direct association of CK2α proximal to transcriptional start sites (TSS) of active loci in the genome. Those included histone genes on chromosome 6, identifying CK2 as an important mediator of gene transcription that may enable cells to advance in the cell cycle.

# Results

## Nuclear localization of CK2α after in vivo phosphorylation contributes to normal cell proliferation and to its catalytic activity

Because functions of nuclear CK2 in cellular processes, such as proliferation in oncogenesis, are largely unknown, we examined subcellular localization of CK2 in relation to cell cycle progression. CK2 activity in the nucleus increased ~3x as human normal fibroblast cells synchronously progressed through the cell cycle, nearly platea duing 6 h after serum stimulation (Fig 1A, *left*), and as generally reported, in HeLa cells, cellular CK2 contents were up-regulated in the nuclear fraction compared with those in the cytosol (Fig 1A, *right*). In 2D PAGE, anti-CK2α antibodies reflected protein spots with decreased $pI$ (Fig 1B). Metabolic $^{32}$P-labeling of synchronously growing cells confirmed the up-regulation of phosphorylation in anti-CK2α immune-precipitates. Autoradiographic gels of proteins after metabolic $^{32}$P[Pi]-labeling showed significantly higher levels of phosphoproteins associated with CK2α complexes, with much higher levels in the nuclear fraction (Fig 1C, *right*) than in the cytosol after growth stimulation (Fig 1C, *left*), as was also seen in confocal images of normal epithelial RPE cells (Fig 1D).

Immunoprecipitated CK2α from cells that progressed synchronously through $G_1$ for 6 h was excised from SDS–PAGE gels, followed by mass spectrometry, and verified by detection of multiple phosphopeptides derived from CK2α, in which three residues phosphorylated in vivo, Ser 7, Ser 194, and Ser 287, were identified (Figs S1A and S2B). These results raised the question of whether in vivo phosphorylation of CK2α might contribute to its nuclear localization and activity. We tested this hypothesis by expressing FLAG-tagged, WT CK2α, and alanine mutants of these serines in cultured cells. A significant loss of nuclear localization was

associated with decreased enzymatic activity in cells bearing recombinant S7A mutant protein, and a partial loss of translocation and activity was also observed with S194A and S287A (Fig 1E). Based on these results, we hypothesized that in vivo phosphorylation at Ser 7 in CK2α may be important to up-regulate enzymatic activity associated with its nuclear localization for cell proliferation. CRISPR-Cas9n was used to create *CK2α* knock-out in RPE cells (CK2-ko) to use as controls. These were devoid of CK2 catalytic subunit α (Fig S1C), and exhibited cell proliferation decreased by 5–22% 6 d after serum stimulation (Fig 1E). A growth assay using CK2-ko expressing wt CK2α, and CK2α with a phosphorylation site mutation, S7A, demonstrated that cells expressing S7A CK2 showed significantly impaired growth compared with cells expressing wt CK2α. The number of cells decreased 15% by the 4th d (Fig 1F). These results demonstrate that loss of nuclear translocation correlates well with the loss of proliferative activity, which suggest the importance of in vivo phosphorylation of CK2α for enzymatic activity, translocation into the nucleus, and cell proliferation.

## Functions of proteins associated with nuclear CK2

To investigate the physiological significance of nuclear CK2α, we then analyzed CK2-immune complexes to identify proteins associated with CK2α in synchronized cells, in either early or late $G_1$ after growth stimulation. We purposely avoided drug-induced synchronization in $G_1$ to exclude experimental artifacts caused by reagents, such as perturbation of protein–protein interactions or other physiological processes. Protein lists from late $G_1$ cells were classified by their molecular function or biological process (Fig 2A), and numbers of identified proteins from early or late $G_1$ cell populations are shown in a Venn diagram (Fig 2B). To estimate relative protein abundances of CK2-interacting proteins, we evaluated these two sets of proteins using the emPAI algorithm (Shinoda et al, 2010). Total spectrum counts were similar regarding average emPAI values and the number of identified proteins (Fig 2C and D). Gene ontology classifications (Fig 2E) depict similar functions between early and late $G_1$ phases, such as RNA-post transcriptional modifications ($P = 1.57$–$2.62 \times 10^{-9}$) and gene expression ($P = 3.10$–$9.97 \times 10^{-7}$) during $G_1$, with similar $P$-values between early and late $G_1$. However, protein signatures in the two growth phases were largely different (Table 1) (original lists in Tables S1 and S2). These functional classifications demonstrated chromatin proteins, transcription factors, RNA-splicing factors, and epigenetic chromatin modifiers that were associated with nuclear CK2 during progression through $G_1$. Of particular interest was *assembly of RNA polymerase II complex* in the canonical pathway, which appeared only in late $G_1$ ($P = 7.26 \times 10^{-8}$). As constituents of RNA pol II, multiple peptides derived from RPB4 and RPB3 were detected in nuclear CK2 complexes from late $G_1$ cells. We identified constituents of nuclear CK2 complexes that are closely related to gene expression, such as histone proteins, which constitute nucleosomes, and transcription factors, and we compared them between early and late $G_1$ (Tables 2 and 3). These results clearly suggest that nuclear CK2 complexes are not consistent and depend on cell cycle progression. Collectively, these findings comprise the first piece of evidence for altered interactions of CK2 with nuclear proteins linked to cell proliferation. These observations motivated

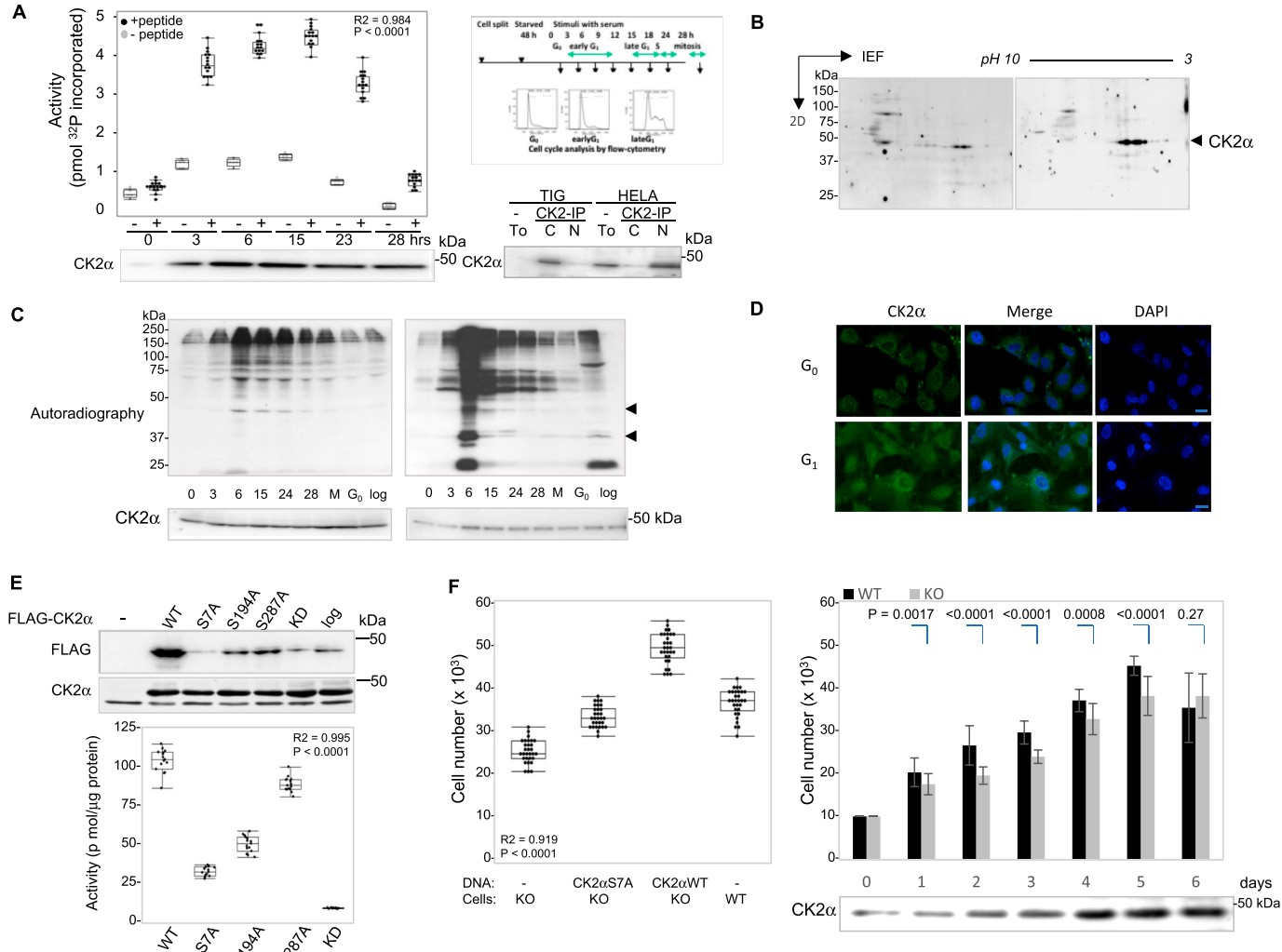

**Figure 1. Activation and nuclear localization of CK2 depends on the proliferative phase of the cell cycle.**
**(A)** Elevation of CK2 activity in the nuclear fraction during cell cycle progression (left). CK2 activity in anti-CK2α immunopreocipitates was measured against the substrate peptide (black), or in the absence of the substrate peptide (gray). Data represent the median, and boxes show interquartile ranges from each (n = 5) of three independent experiments. Differences were statistically analyzed with one-way ANOVA with repeated measures and Tukey's multiple comparison test. *Lower left*, protein contents of CK2α by Western blotting. *Upper right*, the time course of experiments for harvesting cells, and images from flow cytometry using propidium iodide for labeling DNA. Representative images from two independent experiments. *Lower right*, CK2 protein levels are compared between normal (TIG-7) and human cancer cells (HeLa) in which cell fractionation followed by immunoprecipitation with anti-CK2 antibodies were performed. Representative images are shown from three independent experiments. **(B)** Anti-CK2α immunoprecipitated proteins from nuclear fractions were separated by 2D-gel electrophoresis, transferred to membranes and probed with anti-CK2α antibodies. Representative results from TIG-7 cells in early G₁ (*left*) and late G₁ (*right*) are shown. An arrowhead indicates the position of CK2α. The images shown are representative 2D gels from two independent experiments. **(C)** In vivo $^{32}P_i$-labeling of TIG-7 cells followed by cell fractionation into cytosolic (*left*) or nuclear fractions (*right*) and anti-CK2α immunoprecipitation. Representative autoradiographic images of SDS–PAGE gels are shown from two independent experiments. **(D)** Images of endogenous CK2α protein. Human monolayer-cultured RPE cells, arrested in either G₀ or early G₁, were stained with anti-CK2α followed by FITC-labeled secondary antibody. Cells were also stained with DAPI to visualize DNA. Scale bar: 20 μm. Representative images are shown from three independent experiments. **(E)** Decreased nuclear localization of CK2α harboring phosphorylation site mutations. RPE cells expressing WT, phosphorylation site mutants or kinase-dead (kd) mutant CK2α, tagged with FLAG epitope as indicated, were serum-starved, and synchronously grown to early G₁, except cells in logarithmically growing cells. Nuclear proteins were immunoprecipitated with anti-FLAG antibody and probed with monoclonal anti-FLAG antibody (*upper panel*). Representative images are shown from three independent experiments. CK2α proteins in total lysates were probed with anti-CK2α antibody (*middle panel*). CK2 activity toward substrate peptides is compared between recombinant, wt, corresponding phosphorylation-site mutants and kd mutants of CK2α (*lower panel*). Data represent the median, and boxes show interquartile ranges from each (n = 3) of three independent experiments. Differences were analyzed with one-way ANOVA with repeated measures and Tukey's multiple comparison test. **(F)** Decreased cell proliferation in CK2α-depleted clones (CK2-ko) compared with wt cells. (*Left*) CK2-ko cells expressing FLAG-tagged CK2α-Ser 7 substituted to alanine (S7A), or CK2α wt were split as described, and the number of cells on each triple plate was counted at Day 4, as shown at the *right*. Data represent the median, and boxes show interquartile ranges from each (n = 5) of five independent experiments. Differences were analyzed with one-way ANOVA with repeated measures and Tukey's multiple comparison test. (*Right*) 1 × 10⁴ of RPE or CK2-ko cells were split into 12-well culture plates at time zero. The number of cells in each triple plate was counted every 24 h for 6 d after stimulation. Data represent the means ± SDs of three independent experiments. Differences were analyzed with one-way ANOVA with repeated measures and Tukey's multiple comparison test.
Source data are available for this figure.

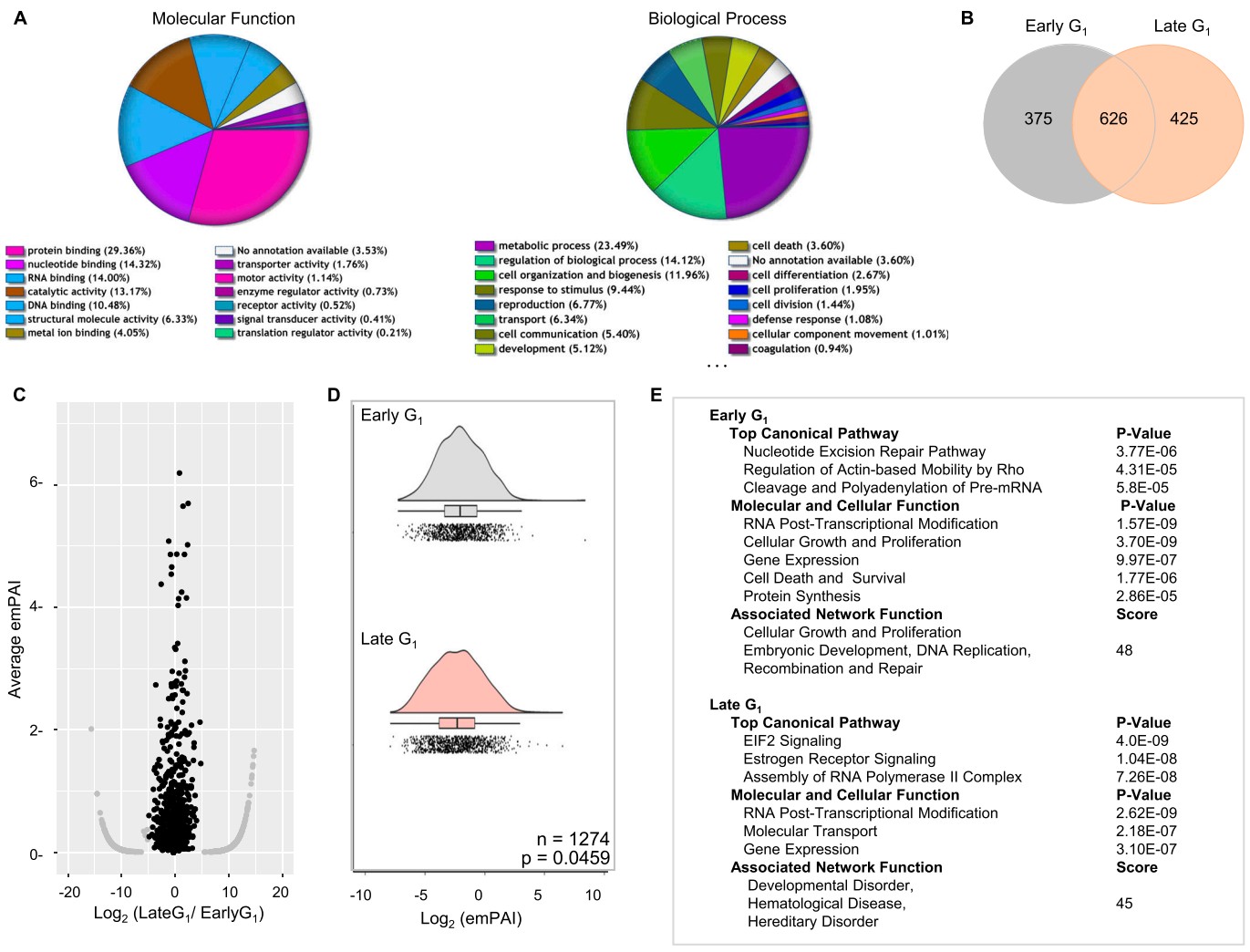

**Figure 2. Assessing components of the CK2 complex in the nucleus demonstrate a role in gene expression.**
**(A)** Functional annotation of CK2-associated proteins in the late $G_1$ phase. The pie chart shows the distribution of the molecular function (*left*) or biological process (*right*) among proteins identified in this study using Gene Ontology analysis. **(B)** Venn diagram showing overlap between proteins identified by mass spectrometry analysis from early or late $G_1$ cells. **(C)** Volcano plots of nuclear CK2 complexes between early $G_1$ (n = 885) versus late $G_1$ (n = 989). Average emPAI values are plotted as a function of the ratio (log base 2), which shows a similar distribution of identified proteins. **(D)** Raincloud plots shows proteins identified as a function of $\log_2$ emPAI ratios for early versus late $G_1$. Statistical analyses were performed with Wilcoxon signed-rank test. **(E)** Summary of IPA analysis in regard to top canonical pathways or molecular and cellular functions of nuclear CK2 complexes in the two cell populations.

us to examine a potential role of nuclear CK2 in controlling transcriptional networks.

### CK2–ChIP-seq demonstrated CK2's association with active gene loci

To further assess the role of nuclear CK2$\alpha$ in transcription during the cell cycle, we performed ChIP-seq analysis, examining the association of CK2 with fragmented chromatin during early or late $G_1$ (Fig 3A, *upper*). The genome-wide profile of CK2-chromatin interactions was determined using ChIP-seq peak call (false discovery rate [FDR] < 0.001), which yielded 9,031 unique peaks in wt cells that significantly decreased in CK2-ko cells (Fig 3A, *lower*). This analysis revealed more significant enrichment of CK2 binding in late $G_1$ than in early $G_1$, with the latter showing 4,240 binding sites.

Distribution of CK2 ChIP-seq peaks in late $G_1$ cells over known gene annotations were as follows: upstream promoters (3,667, 40.6%), genic regions (3,479, 38.5%), downstream (514, 5.7%) and intergenic regions (1,371, 15.2%), most of these with narrow peaks within 2 kb of coding genes (Fig 3B). Average read density of CK2–ChIP-seq signal profiles was concentrated mostly from 1,000 bp upstream to 1,000 bp downstream of TSS (Fig 3B, *left*, *right*). Gene Ontology classification of gene lists extracted from CK2–ChIP-seq reads from late $G_1$ cells, indicated functional involvement of CK2 in *gene expression* with *P*-values as low as $1.13 \times 10^{-53}$ (Fig 3C), suggesting association of CK2 with Pol II promoters. In order to investigate this further, we conducted pulldown assays with CK2 in nuclear extracts. Fig 3D shows that RNA pol II indeed co-immunoprecipitated with anti-CK2 antibody in $G_1$ phase cells expressing wt CK2, but not in CK2-ko cells in which CK2$\alpha$ was depleted. This result confirmed

**Table 1.** Functional classification of protein signatures associated with nuclear CK2α.

| Functional classifications | | | | | | | |
|---|---|---|---|---|---|---|---|
| cell cycle in $G_1$ | accession | description | score | coverage | # peptides | MW [kD] | calc. pI |
| Histone proteins | | | | | | | |
| Early | Q16777 | Histone H2A type 2-C | 20478.93 | 55.04 | 5 | 14.0 | 10.90 |
| | P62805 | Histone H4 | 19401.40 | 58.25 | 12 | 11.4 | 11.36 |
| Late | P16403 | Histone H1.2 | 32093.90 | 35.21 | 12 | 21.4 | 10.93 |
| | P10412 | Histone H1.4 | 31737.32 | 33.33 | 10 | 21.9 | 11.03 |
| Transcription factors | | | | | | | |
| Early | Q9NYF8 | Bcl-2-associated transcription factor 1 | 1824.83 | 36.30 | 29 | 106.1 | 9.98 |
| | O00268 | Transcription initiation factor TFIID subunit 4 | 791.65 | 18.80 | 13 | 110.0 | 9.94 |
| Late | O00268 | Transcription initiation factor TFIID subunit 4 | 2501.42 | 20.18 | 17 | 110.0 | 9.94 |
| | Q8WXI9 | Transcriptional repressor p66-beta | 1937.69 | 37.27 | 13 | 65.2 | 9.70 |
| Methylation | | | | | | | |
| Early | O14744 | Protein arginine N-methyltransferase 5 | 948.39 | 44.74 | 17 | 72.6 | 6.29 |
| | Q9BQA1 | Methylosome protein 50 (WDR77) | 630.17 | 37.13 | 6 | 36.7 | 5.17 |
| Late | P22087 | rRNA 2'-O-methyltransferase fibrillarin | 1879.01 | 52.02 | 12 | 33.8 | 10.18 |
| | O95983 | Methyl-CpG-binding domain protein 3 (MBD3) | 1769.49 | 26.46 | 7 | 32.8 | 5.34 |
| Repair proteins | | | | | | | |
| Early | P12956 | X-ray repair cross-complementing protein 6 | 2626.15 | 46.96 | 26 | 69.8 | 6.64 |
| | P13010 | X-ray repair cross-complementing protein 5 | 1752.65 | 53.96 | 24 | 82.7 | 5.81 |
| Late | Q9Y230 | RuvB-like 2 | 1370.12 | 37.37 | 12 | 51.1 | 5.64 |
| | Q9Y265 | RuvB-like 1 | 1249.12 | 30.04 | 10 | 50.2 | 6.42 |
| RNA helicases | | | | | | | |
| Early | O75643 | U5 small nuclear ribonucleoprotein 200-kD helicase | 3349.71 | 34.97 | 52 | 244.4 | 6.06 |
| | Q08211 | ATP-dependent RNA helicase A | 2466.37 | 31.42 | 27 | 140.9 | 6.84 |
| Late | O00571 | ATP-dependent RNA helicase DDX3X | 1340.82 | 31.42 | 15 | 73.2 | 7.18 |
| | P60842 | Eukaryotic initiation factor 4A-I | 1266.42 | 39.66 | 12 | 46.1 | 5.48 |
| Splicing facotrs | | | | | | | |
| Early | Q6P2Q9 | Pre-mRNA-processing splicing factor 8 | 2229.28 | 28.18 | 43 | 273.4 | 8.84 |
| | O43143 | Putative pre-mRNA splicing factor ATP-dependent RNA helicase DHX15 | 944.91 | 14.84 | 9 | 90.9 | 7.46 |
| Late | O75533 | Splicing factor 3B subunit 1 | 1225.81 | 26.69 | 19 | 145.7 | 7.09 |
| | Q13435 | Splicing factor 3B subunit 2 | 1198.62 | 21.34 | 14 | 100.2 | 5.67 |
| Acetylation | | | | | | | |
| Early | Q92769 | Histone deacetylase 2 (HDAC2) | 553.95 | 18.85 | 6 | 55.3 | 5.91 |
| | O00422 | Histone deacetylase complex subunit SAP18 | 401.53 | 40.52 | 4 | 17.5 | 9.35 |
| Late | O00422 | Histone deacetylase complex subunit SAP18 | 1184.14 | 41.18 | 6 | 17.5 | 9.35 |
| | Q92769 | Histone deacetylase 2 (HDAC2) | 503.66 | 17.21 | 5 | 55.3 | 5.91 |
| Chromatin remodeling factors | | | | | | | |
| Early | P11388 | DNA topoisomerase 2-alpha | 1726.00 | 25.47 | 29 | 174.3 | 8.72 |
| | Q02880 | DNA topoisomerase 2-beta | 897.24 | 12.48 | 17 | 183.2 | 8.00 |
| Late | Q969G3 | SWI/SNF-related matrix-associated actin-dependent regulator of chromatin subfamily E member 1 | 1168.91 | 30.41 | 9 | 46.6 | 4.88 |
| | Q5SSJ5 | Heterochromatin protein 1-binding protein 3 | 1139.98 | 33.45 | 17 | 61.2 | 9.67 |

**Table 1. Continued**

| Functional classifications | | | | | | | |
|---|---|---|---|---|---|---|---|
| cell cycle in $G_1$ | accession | description | score | coverage | # peptides | MW [kD] | calc. pI |
| Transcriptional machineries | | | | | | | |
| Early | P19388 | DNA-directed RNA polymerases I, II, and III subunit RPABC1 | 282.98 | 40.00 | 6 | 24.5 | 5.95 |
| | P24928 | DNA-directed RNA polymerase II subunit RPB1 | 173.74 | 2.13 | 2 | 217.0 | 7.37 |
| Late | O15446 | DNA-directed RNA polymerase I subunit RPA34 | 969.98 | 20.39 | 4 | 55.0 | 8.51 |
| | P19388 | DNA-directed RNA polymerases I, II, and III subunit RPABC1 | 295.05 | 41.43 | 6 | 24.5 | 5.95 |
| Hisotne chaperones | | | | | | | |
| Early | Q9Y5B9 | FACT complex subunit SPT16 | 1182.54 | 27.03 | 21 | 119.8 | 5.66 |
| | Q08945 | FACT complex subunit SSRP1 | 641.97 | 17.91 | 9 | 81.0 | 6.87 |
| Late | Q09028 | Histone-binding protein RBBP4 | 944.41 | 31.06 | 10 | 47.6 | 4.89 |
| | Q13112 | Chromatin assembly factor 1 subunit B | 89.32 | 4.65 | 2 | 61.5 | 7.50 |
| Replication | | | | | | | |
| Early | P27694 | Replication protein A 70 kD DNA-binding subunit | 523.96 | 19.64 | 8 | 68.1 | 7.21 |
| | P35244 | Replication protein A 14 kD subunit | 229.61 | 41.32 | 3 | 13.6 | 5.08 |
| Late | P35249 | Replication factor C subunit 4 | 676.00 | 30.58 | 7 | 39.7 | 8.02 |
| | P35251 | Replication factor C subunit 1 | 464.56 | 5.05 | 4 | 128.2 | 9.36 |

CK2-associated proteins were identified by nano-LC mass spectrometry. The two most highly expressed proteins for each of the early $G_1$ and late $G_1$ cells are shown, according to their Mascot scores, classified into 11 categories regarding chromatin dynamics and nuclear functions. Protein coverages (%), numbers of peptides detected, molecular weights (kD), and calculated pH values are indicated. All proteins identified in the CK2-immune-complexes are listed in Tables S1 and S2, from early $G_1$ and late $G_1$ cells, respectively.

**Table 2. Cell cycle-dependent enrichment of histone proteins in the nuclear CK2α complexes.**

| Accession | Description | Score (cell cycle in $G_1$) | |
|---|---|---|---|
| | | early | late |
| P16403 | Histone H1.2 | 3376.38 | 32093.9 |
| P10412 | Histone H1.4 | 3269.68 | 31737.32 |
| P16402 | Histone H1.3 | | 31633.01 |
| Q16777 | Histone H2A type 2-C | 20478.93 | 19368.15 |
| O60814 | Histone H2B type 1-K | 11338 | 15504.75 |
| P06899 | Histone H2B type 1-J | 10945.56 | 15461.91 |
| P0C0S5 | Histone H2A.Z | 7256.44 | 10520.64 |
| Q8IUE6 | Histone H2A type 2-B | 7402.83 | 9127.07 |
| P07305 | Histone H1.0 | | 648.05 |
| Q71DI3 | Histone H3.2 | | 414.05 |
| P62805 | Histone H4 | 19401.4 | |
| P16104 | Histone H2A.x | 8524.62 | |
| P84243 | Histone H3.3 | 1724.7 | |

Protein lists in each category were extracted, according to their Mascot scores, from the list of nuclear proteins interacting with CK2, as shown in Tables S1 and S2. Accession numbers, protein descriptions, and Mascot scores are indicated.

binding of RNA pol II with nuclear CK2 complexes during progression of the $G_1$ phase. Consensus motif sequences for CK2-bound loci in the genome were extracted from ChIP-seq data,

demonstrating primary sequences shared with general transcription factors, such as nuclear transcription factor Y-β (Fig 3E). Identified motifs are similar to a "CAT box" which is a consensus sequence upstream to transcription sites and to potential pol II binding sites. Interestingly, motifs shared with SP1 transcription factor, were also identified by de novo motif analysis (Fig 3E).

To address whether genomic occupancy of CK2 is related to gene expression, we analyzed ChIP-seq data from late $G_1$ cells by calculating the number of reads of broad peaks per 10 kb. Logarithmic coverage of CK2–ChIP-seq fragments was closely correlated with those of RNA polymerase II (Pol II)–ChIP-seq data (Fig S2A). In addition, GO analysis demonstrated functional homology in gene regions covered by Pol II and CK2 ChIP experiments, with enrichment in terms for mRNA catabolic process and splicing (Fig S2B) (Yu et al, 2012). Enrichment of CK2, Pol II, Pol II-phospho-Ser 2, and H3K4me3, from the TSS to the TES of all protein-coding genes, delineated similar profiles of CK2 and Pol II, which interact with chromatin at active promoters (Fig 3F). Pearson correlation of read counts from our ChIP-seq signal profiles in combination with publicly available datasets (Ji et al, 2015), showed that CK2 has similar coverage to Pol II and H3K4me3 throughout the entire genome (Fig 3G). In conclusion, these results suggest that CK2 is recruited to promoter regions associated with transcriptional activity. As supported by proteomic data (Fig 2E), and by anti-CK2 immunoprecipitation, CK2 recruitment may be driven partially by interactions with Pol II.

We asked whether CK2 participates critically in gene expression during late $G_1$, by considering the finding that more genomic regions for CK2 binding were identified in late $G_1$ than in early $G_1$. In

**Table 3. Cell cycle–dependent enrichment of transcriptional regulators in the nuclear CK2α complexes.**

| Accession | Description | Score (cell cycle in $G_1$) | |
|---|---|---|---|
| | | early | late |
| O00268 | Transcription initiation factor TFIID subunit 4 | 791.65 | 2501.42 |
| Q8WXI9 | Transcriptional repressor p66-beta | | 1937.69 |
| Q86YP4 | Transcriptional repressor p66-alpha | | 1294.11 |
| O75531 | Barrier-to-autointegration factor | | 927.86 |
| P78347 | General transcription factor II-I | | 722.47 |
| P51532 | Transcription activator BRG1 | | 623.91 |
| Q14683 | Structural maintenance of chromosomes protein 1A | 440.33 | 589.02 |
| Q13263 | Transcription intermediary factor 1-beta | 685.37 | 455.86 |
| Q9UKN8 | General transcription factor 3C polypeptide 4 | | 397.45 |
| Q6KC79 | Nipped-B-like protein | | 373.33 |
| Q9NYF8 | Bcl-2-associated transcription factor 1 | 1824.83 | |
| Q14676 | Mediator of DNA damage checkpoint protein 1 (MDC1) | 531.6 | |
| P46100 | Transcriptional regulator ATRX | 447.01 | |
| P17480 | Nucleolar transcription factor 1 | 384.8 | |
| Q92804 | TATA-binding protein-associated factor 2N (TAF15) | 383.99 | |
| P28370 | Probable global transcription activator SNF2L1 | 334.83 | |
| Q15059 | Bromodomain-containing protein 3 | 305.96 | |

Protein lists in each category were extracted, according to their Mascot scores, from the list of nuclear proteins interacting with CK2, as shown in Tables S1 and S2. Accession numbers, protein descriptions, and Mascot scores are indicated.

general, activation of histone genes is linked to DNA synthesis, followed by newly formed nucleosomes with DNA duplication in the S phase. As we expected, ChIP-seq profiles revealed CK2 enrichment on histone gene clusters on chromosome 6 (Fig 4A), with high $\log_{10}$ P-val/Input, for *H2AK*, *H2AC*, *H2AH*, *H2BK*, *H4A*, and *H4H* loci (Fig 4B), and confirmed in independent CK2-ChIP–qPCR experiments (Fig 4C). These results demonstrated CK2α enrichment at histone gene loci on active chromatin where trimethylated histone H3K4, monoacetylated histone H3K27, and RNA Pol II were similarly recruited. On the other hand, RNA levels of typical histone genes, including *H1C*, *H2AC*, *H2BJ*, *H3H*, and *H4B*, were down-regulated in CK2-ko cells in late $G_1$ (Fig S3A). In order to address the impact of CK2 phosphorylation on association with chromatin and gene transcription, CK2-ChIP was examined for quantitative loading of CK2-wt compared with CK2-mutant proteins harboring a phosphorylation site mutation or a kinase-dead mutation, onto a *HIST1H2AC* locus using the reconstituting system with CK2-ko cells. Expression of CK2α with phosphorylation site mutation with S7A could not rescue the reduced association of CK2 with chromatin loci, and the 60% lower expression of the *HIST1H2AC* gene seen in CK2-ko cells, which was fully rescued in CK2-wt–expressing cells (Fig 4D). Likewise, kinase-dead mutants showed 30% less *HISTH2AC* gene expression than CK2-wt. Together, this suggests that phosphorylation of CK2α and its enzymatic activity are required for its genomic occupancy.

In order to address target genes for CK2 recruitment, we analyzed the transcriptomes of RPE wt and CK2-ko cells, both arrested in late $G_1$ during progression of the cell cycle. These results demonstrated 303 genes that either increased or decreased more than 1.5-fold

(Fig 5A, full list in Table S3). When these genes were compared with data obtained by CK2-ChIP-seq, 104 genes were shared between the two experiments (Fig 5B and C; full list in Table S4). Here, several growth-associated genes were up-regulated, including RNA helicase (HELLS), PRR11, CDCA8, and DNA topoisomerase 2 (TOP2A), HMGB2, EIF2B4, and CDKN3, which are thought to serve important functions during growth stimulation, many of which are known to be up-regulated in various cancers. Sequence tag-mapping revealed that CK2α was significantly enriched at genomic loci for these genes (Fig 5B). This was further confirmed by ChIP–qPCR analysis (Fig 5D, *upper*), with many genes demonstrating cell cycle-dependent up-regulation in late $G_1$ (Fig 5D, *lower*). Overall, these results document CK2 recruitment to promoter-proximal regions of growth-associated genes after serum exposure of quiescent cells, some of which are target genes for CK2 recruitment.

In regard to histone proteins, qPCR by total RNA (Fig S3A) and Western blotting experiments (Fig S3B) both showed decreases in histone gene expression in CK2-ko RPE cells compared with RPE wt conditions. Interestingly, protein levels of linker histone H1, but not histones H3 or H4, were down-regulated in CK2-ko cells, and association of H1 with CK2 occurred predominantly in nuclei of late $G_1$ cells, as demonstrated by CK2 immunoprecipitation. Therefore, cellular imaging was performed using late $G_1$ cells stained with anti-histone H1, which confirmed decreased levels of H1 protein in the nuclear fraction of CK2-ko cells, as observed by 3D-localization analysis of H1 protein in the videoclip (Fig S3C, Supplemental Data 1). Although we have not addressed the mechanistic link of CK2 recruitment to genomic

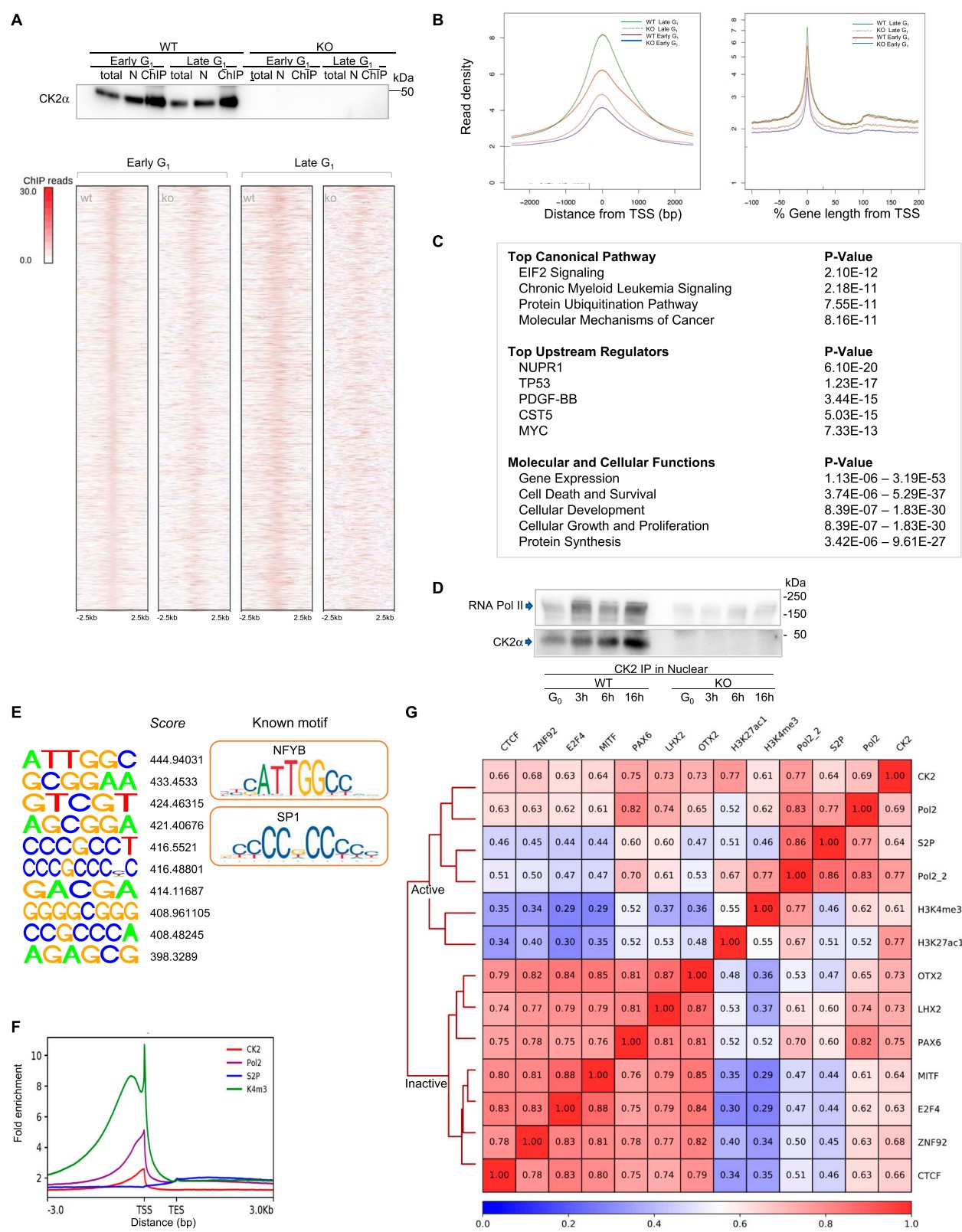

**Figure 3. Chromatin immunoprecipitation-sequencing (ChIP-seq) profiles of CK2α demonstrate localization of CK2α at active genes.**
**(A)** Preparation of CK2α-ChIP followed by Western blotting with anti-CK2α antibody (*upper*). Representative images are shown from three independent experiments. Heatmaps of CK2α-binding profiles around peaks of CK2-wt cells and CK2-ko cells (from 2.5 kb upstream of the transcriptional start sites [TSS] to 2.5 kb downstream of peak summits) in early or late G$_1$(*lower*). Gene lengths are scaled to the same size. Increased numbers of CK2α interactions with the whole genome in late G$_1$ versus early G$_1$

loci, these results suggest involvement of CK2 in gene expression to provide essential components for cell proliferation.

### CK2 associates with actively transcribed ribosomal genes during rRNA synthesis

Our recent finding that nuclear and nucleolar CK2α localization might be statistically useful for predicting adverse outcomes of patients with invasive ductal breast carcinomas (Homma et al., 2021, 2022), prompted us to examine whether CK2 is also recruited to *rDNA* genes localized in the nucleolus. Results of CK2-ChIP–qPCR show that expression of 5′ETS pre-rRNAs is significantly down-regulated in CK2-ko cells compared with that in CK2-wt cells (Fig 6A, *left*). Similar results were obtained using three additional sets of *rDNA* primers. As the next step, we investigated cellular components potentially associated with CK2 to promote rRNA expression. Notably, multiple RNA helicases were extracted from nuclear CK2α complexes (Table 1 and Fig 6B), which are also localized in the nucleolus. As previous studies have demonstrated that DEAD-box RNA helicase DDX21 may sense the transcriptional status of both RNA polymerase II (Pol II) and I (Pol I) in human cells (Calo et al, 2015), CK2–ChIP-seq reads from RPE-wt cells were compared with those from genome browser tracks displaying DDX21–ChIP-seq signal in HEK293 cells. CK2 broadly associated with transcribed regions of rRNA, in a pattern quite similar to that of DDX21 (a profile characteristic of Pol I-associated transcriptional regulators, Calo et al, 2015) (Fig 6A, *right*). To explore this, we evaluated proteins associated with nuclear CK2α, using immunoprecipitation of cytosolic and nuclear fractions from cells expressing each of the newly cloned RNA helicases, DDX3X, DDX9, DDX21, EIF4A1, and also Pol I-associated transcription factors, RRN3/TIF-1A, TAF1A, and SP1 (Fig 3E), which are involved in assembly of pre-initiation complexes during Pol I-dependent transcription. Those components were evaluated mostly in nuclear CK2 complexes, along with each nuclear or nucleolar protein (Fig 6B and C), such as nucleolar GTP-binding protein 1 and NOLC1. These results demonstrate for the first time that CK2 may facilitate ribosome biogenesis by combinatorial association with multifaceted nuclear and nucleolar components.

## Discussion

The present study demonstrated activation of nuclear CK2 during cell cycle progression in a manner that is closely associated with its nuclear localization and enhanced phosphorylation levels, both of which are likely essential for cell proliferation. Identification of CK2 complexes to explore the nuclear function of CK2 revealed a group of molecules involved in epigenetic chromatin remodeling and gene expression. CK2–ChIP-seq analysis demonstrated that CK2 is recruited to active genomic regions, proximal to the TSS, and that CK2 exhibited a profile quite similar to that of Pol II-binding sites. Furthermore, as many nucleolar proteins were found among nuclear CK2-associated molecules, we have now shown by both ChIP-seq and ChIP–qPCR analysis that CK2 binds to *rDNA* gene regions in the nucleolus and is involved in rRNA expression in normal cells during progression of the cell cycle.

Nuclear translocation of CK2 is accompanied by phosphorylation of CK2α. CK2 mutants depleted of individual phosphorylation sites revealed that phosphorylation of CK2α may be involved in translocation of CK2α into the nucleus. This finding and other results involving recombinant proteins further demonstrate the importance of CK2α phosphorylation for its activation, localization into the nucleus, and for cell proliferation. The consensus sequence of these phosphorylation sites searched in NetPhos predicted each upstream kinase with top score: PKCα for S7, MAP2K6 for S194, and CK2 itself for S287. Future studies will clarify which upstream kinase is responsible for phosphorylation in vivo. These results demonstrate, for the first time, the functional importance of phosphorylated CK2α in vivo for cell proliferation.

Classification and comparative estimation of protein abundance in CK2 complexes between early and late $G_1$ suggest participation of CK2 in dynamic gene activation. A link between CK2 and chromatin remodeling was previously shown by ChIP experiments in which CK2 and transcription factor PC4 associate with the downstream promoter element-dependent *IRF-1* promoter for its transcription (Lewis et al, 2005), followed by reports on association of CK2 with active genes in mammalian cells and yeast (Panova et al, 2006; Basnet et al, 2014). Therefore, we sought to delineate genome-wide binding of CK2 to actively transcribed genes with deep-sequencing of cells which synchronously progressed throughout the cell cycle. Higher peak calling of CK2 was observed in late $G_1$ than in early $G_1$, and our ChIP-seq profiles of CK2 during progression of the cell cycle confirmed the reliability of the CK2 distribution profile using independent CK2-ChIP–qPCR experiments. On the other hand, numbers of chromatin-binding sites for epigenetic markers (trimethylated histone H3K4, acetylated H3K27, RNA polymerase II, or RNA polymerase II with phosphorylated Ser 2), were quite similar during progression through the $G_1$ phase (Accession No GSE226778).

---

cells. Both show much higher levels in CK2-wt than -ko cells. **(B)** Appearance of CK2 ChIP-seq peaks at TSSs with higher read densities in late $G_1$ cells. Distances of CK2α peaks from TSSs (*left*), or whole-gene bodies (right) indicate more than 75% of CK2α binding sites are localized within ±1 kb of a TSS. Gene lengths are scaled to the same size. **(C)** Gene loci for CK2 binding represent pathways primary for "gene expression," and then "cell death and survival," as shown by Ingenuity Pathway Analysis. As seen in the "Molecular and Cellular Functions," "gene expression" was extracted as the most relevant function. **(D)** Association of Pol II with CK2α. Nuclear extracts were immunoprecipitated with anti-CK2α, demonstrating its binding to RNA polymerase II during progression of the cell cycle. Typical results of Western blotting from triplicate experiments are shown. **(E)** De novo motif searches in CK2-bound regions identified potential sites for general transcription factors, such as NFYB and SP1 (*right*). Several identified motifs defined by CK2–ChIP-seq are listed (*left*). **(F)** CK2 enrichment is shown by the red line for all protein-coding genes from 3 kb upstream of the transcriptional start sites to 3 kb downstream, as compared with Pol II (Pol2), Pol II-Ser-P (S2P), and H3K4me3(K4m3). Gene lengths are normalized to the same size. **(G)** Pearson correlation of read counts was produced from our ChIP-seq signal profiles (shown as CK2, Pol2, S2P, H3K4me3, and H3K27ac1) along with publicly available data sets (shown as Pol2_2, OTX2, LHX2, PAX6, MITF, E2F4, ZNF92, and CTCF, Ji et al, 2015) to investigate the degree of similarity between samples. Colors indicate similarity based on coefficients of the log-transformed number of ChIP-seq reads. CK2 demonstrates coverage similar to that of RNA pol II, H3K4me3 and H3K27ac in genomic regions. Source data are available for this figure.

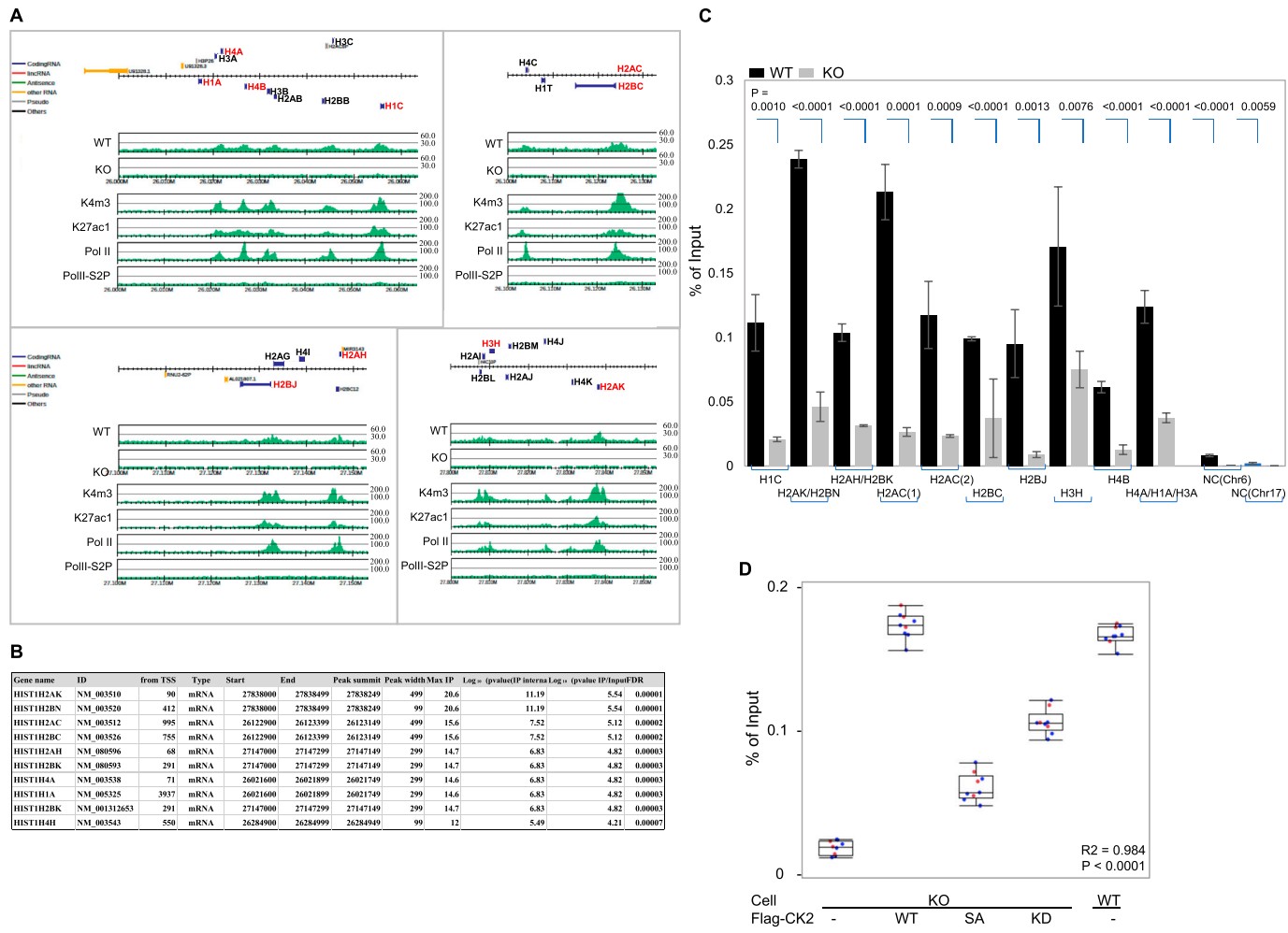

**Figure 4. Chromatin immunoprecipitation-sequencing (ChIP-seq) demonstrates CK2α binding to histone gene loci on Chromosome 6.**
**(A)** ChIP-seq profiles of CK2α, H3K4me3, H3K27ac, PolII, and PolIISer2P obtained from late $G_1$, in CK2α-wt and –depleted cells. Upper boxes show positions of protein-coding genes. Y axes in the six sets of graphs indicate normalized read intensities. **(B)** List of CK2 enrichment peaks in histone gene loci obtained by ChIP-seq. Unique CK2α peaks identified in late $G_1$ WT cells are listed here according to $\log_{10}$ P (IP/input) values. **(C)** Anti-CK2α ChIP–qPCR analysis of histone genes in late $G_1$ WT shows enrichment of CK2α on these histone gene loci, but not on control loci, comparing all with those in CK2α–depleted cells. NC, negative controls on Chromosome 6, and Chromosome 17 as a comparison. Data represent the means ± SD of three independent experiments. Differences were analyzed with one-way ANOVA with repeated measures and Tukey's multiple comparison test. **(D)** Phosphorylation of CK2α is required for recruitment to target gene loci. CK2-ChIP–qPCR demonstrates CK2 binding to *HISTH2AC* loci when CK2 holds phosphorylation sites and catalytic activity. CK2-ko cells were transfected with either CK2α-wt (wt), a phosphorylation site mutant (S7A) or a kinase-dead (kd) construct. Cells were synchronously arrested in $G_0$ and then stimulated with serum for 15 h. Data represent the median, and boxes show interquartile ranges from each (n = 3) of three independent experiments. Differences were statistically analyzed with one-way ANOVA with repeated measures and Tukey's multiple comparison test.
Source data are available for this figure.

As phosphorylation of Pol II at the C-terminus by CK2 has been reported (Trembley et al, 2003), we confirmed the interaction between Pol II and CK2 (Fig 3D). Notably, we found CK2 bound in proximity to active genes at TSSs, including histone genes where transcription levels were elevated during the late $G_1$ phase. By comparing ChIP-seq data with RNA expression analyses of wt and CK2α-ko cells, recruitment of nuclear CK2 to actively transcribed regions was validated. These results suggest that CK2 may be involved in active gene expression in much the same way as Pol II, by tethering basic transcription factors close to the TSS.

According to our CK2–ChIP-seq and ChIP–qPCR analyses, CK2α serves an important function in histone gene transcription during late $G_1$. Also, loci of multiple transcriptional regulators for histone genes (Gokhman et al, 2013), were also enriched in our CK2–ChIP-seq data (Table S5). Most obvious in our proteomic data, appearance of linker histones with high scores in CK2 complexes of late $G_1$, H1.2, H1.4, and H1.3, which interact with linker DNA between two nucleosomes (White et al, 2016), implies that those linker histones may tether CK2 to achieve possible down-regulation of chromatin compaction followed by formation of euchromatin at that site. Association of H2A.z and H2A.x with CK2 along with core histones during the $G_1$ phase suggest that CK2 helps to maintain genomic integrity and stability, because H2A.z and H2A.x have been found in nucleosomes located on both sides of active promoter regions

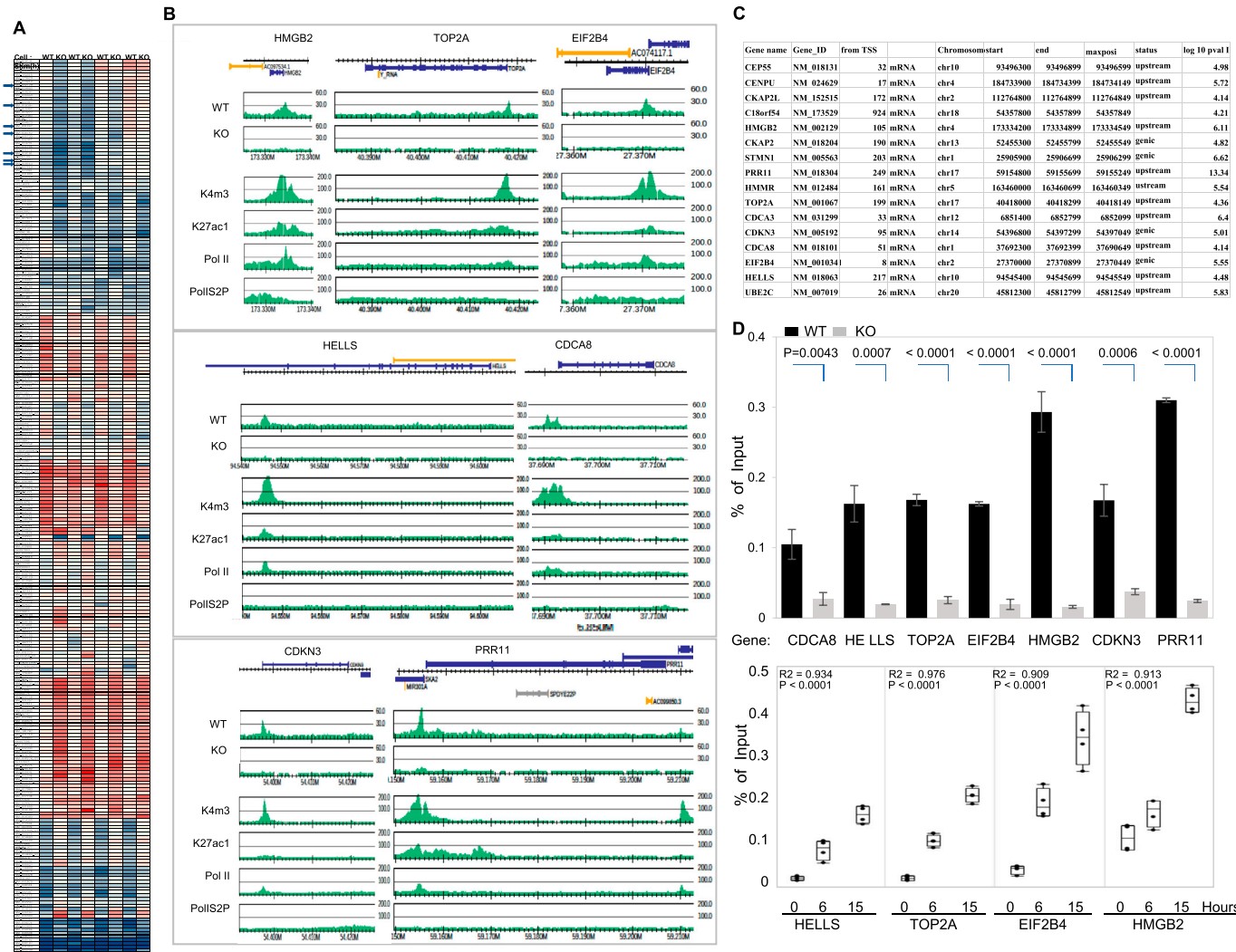

**Figure 5. CK2α is involved in the expression of proliferation-related genes that corresponds to global gene expression analysis.**
**(A)** Heatmap of 303 genes that exhibited significantly altered expression (fold change > 1.5 or < −1.5 and P-value < 0.01) between RPE (wt) and CK2α-depleted RPE cells (CK2-ko) in biological replicates (full list in Table S3). Arrows indicate genes with larger differences that were validated by subsequent analysis. **(B)** Chromatin immunoprecipitation-sequencing (ChIP-seq) profiles of CK2α from late G₁, in both wt and CK2-ko cells. Upper boxes show positions of protein-coding genes. Y axis in the seven sets of graphs indicate normalized read intensities. **(A, C)** Gene lists that were detected both in the expression analysis as in (A), and in the CK2–ChIP-seq data from late G₁ cells, are picked up partially in the order listed in (A) (full list in Table S4). Regions where enrichment was determined by CK2α–ChIP-seq were statistically significant by peak calling, as listed in the table. Gene names, accession codes, length to transcriptional start sites, gene loci, and log₁₀ (P-value IP/Input) are reported. **(D)** Genes validated by CK2–ChIP–qPCR are shown (*upper*), demonstrating CK2α enrichment on growth-associated genes, and down-regulation in CK2α-depleted cells. Data represent means ± SDs of three independent experiments. Differences were analyzed with one-way ANOVA with repeated measures and Tukey's multiple comparison test. (*Lower*) Time-dependent association of CK2 on the *HISTH2AC* locus. CK2α–ChIP–qPCR analysis was carried out using cells synchronously arrested in G₀ and then stimulated at time 0 with serum for 6 or 15 h. CK2α–ChIP–qPCR results with corresponding primers, as indicated. Data represent the median, and boxes show interquartile ranges from each (n = 3) of three independent experiments. Differences were analyzed with one-way ANOVA with repeated measures and Tukey's multiple comparison test.
Source data are available for this figure.

(Olsen et al, 2012; Watanabe et al, 2013). Given that chromatin remodeling enzymes, such as DNA topoisomerase 2 and RNA helicases, which are closely associated with active transcription, displayed the highest scores in early G₁ CK2 complexes, it is likely that CK2 triggers unwinding of nearby chromatin at transcriptionally active sites via energy-consuming activity cooperating with multiple proteins. Overall, our results show that nuclear CK2 complexes are recruited to promoter-proximal regions of histone genes and growth-associated genes after growth factor exposure of quiescent

cells. Recently, increasing evidence for direct association of protein kinases with active gene loci was reported: recruitment of protein kinase C-θ to promoters and transcribed regions of cytokine regulatory genes in human T lymphocytes (Sutcliffe et al, 2011), ERK2 to histone genes, cell cycle-, metabolism-, and pluripotency-associated genes in human ES cells (Goke et al, 2013), EGF receptor tyrosine kinase to chromatin-encompassing transcribed genes, including *EGR1* in HeLa cells (Mikula et al, 2016), and CDK11 to replication-dependent histone genes in HCT116 cells (Gadjuskova

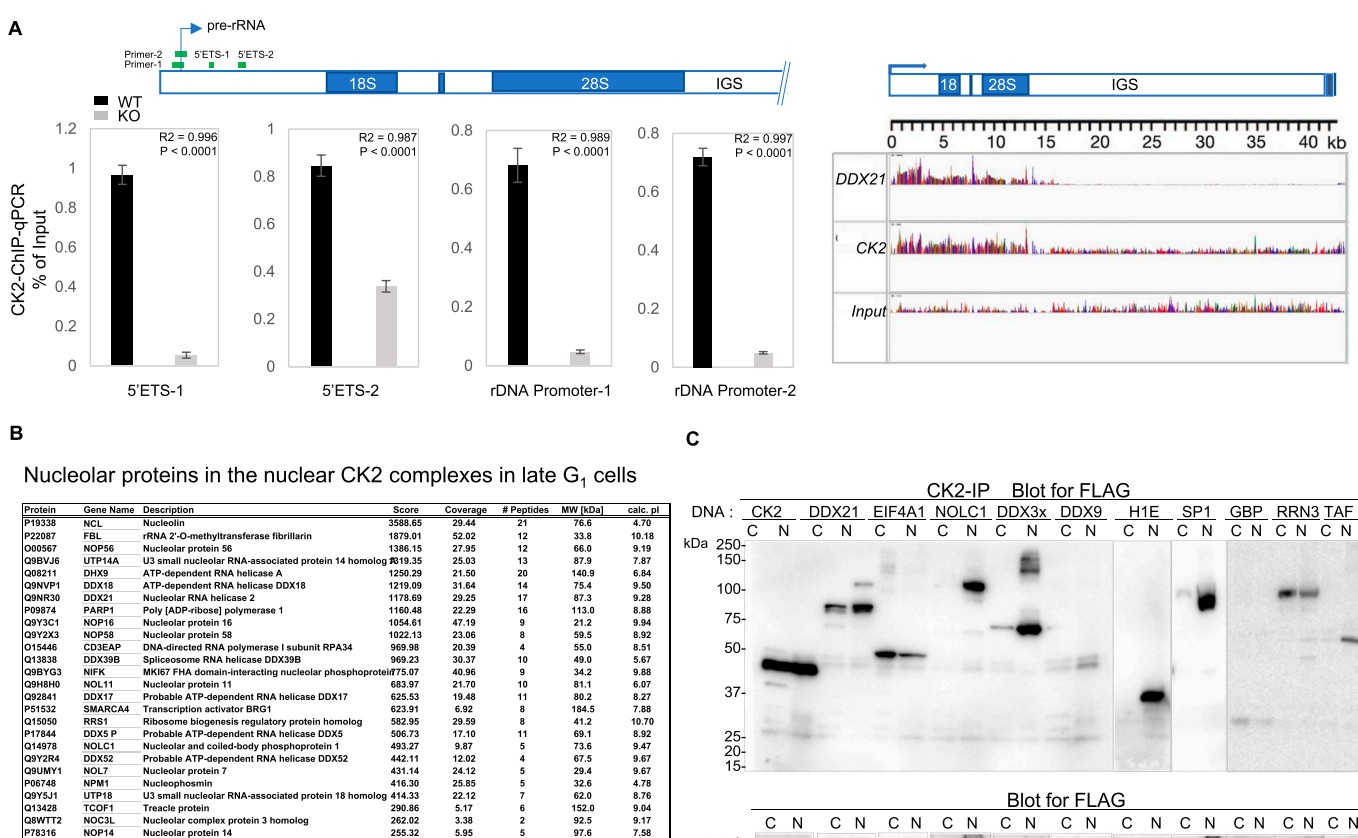

**Figure 6. Recruitment of CK2 to actively transcribed ribosomal genes.**
**(A)** ChIP assay was performed to measure levels of CK2 on *rDNA* promoters and qPCR was performed in triplicate using different primer sets, as shown by blue bars at the top (*left*). Data represent means ± SDs of three independent experiments. Differences were analyzed with one-way ANOVA with repeated measures and Tukey's multiple comparison test. (*Right*) CK2 chromatin immunoprecipitation-sequencing reads were mapped to an annotation file of the human *rDNA* locus and compared with input reads, and also aligned with DDX21 chromatin immunoprecipitation-sequencing signal profiles from publicly available datasets (Calo et al, 2015). IGS, intergenic spacer. **(B)** Nucleolar proteins sorted from nuclear CK2 complexes in late $G_1$ cells (original list in Table S2). Proteins are listed according to Mascot scores. **(C)** Functional association of CK2 with rDNA transcription factors, DNA topology modifiers, and nuclear/nucleolar proteins. Immunoprecipitation with monoclonal anti-CK2 antibody **6A** was performed using the cytosolic or nuclear fraction from HEK293 cells. Each expressed the FLAG epitope-tagged pCDNA3 construct of the indicated gene, and synchronously progressed to late $G_1$ phase in the cell cycle. Representative results of three independent experiments for Western blotting are shown. Blots with total protein from each cell fraction are also shown using anti-FLAG antibody (*lower*). These molecules are categorized as follows: rDNA transcription factors known to associate with ribosomal genes: RRN3/TIF-1A, TAF1A, SP1; DNA topology modifiers: DDX21, EIF4A1, DDX3X, DDX9; nuclear or nucleolar proteins: NOLC1, H1E, nucleolar GTP-binding protein 1.
Source data are available for this figure.

et al, 2020). Although these publications did not address molecular charges because of posttranscriptional modifications by kinases, future studies will clarify mechanistic links that may recruit CK2α to target genomic sites in a locus-specific manner.

We demonstrated that the unphosphorylated form of CK2 is down-regulated in terms of recruitment to and activation of histone gene loci (Fig 4D, *lower*), and in cell proliferation (Fig 1F). Our data clearly demonstrate for the first time that the phosphorylated form of CK2 may participate in transcriptional activation with enhanced activity after phosphorylation and translocation to the nucleus. Phosphorylation may be essential for nuclear localization and also retention of CK2α. These results strongly suggest that phosphorylation-dependent formation of nuclear CK2 complexes is essential for CK2 recruitment to its target genes, which may be required for cell cycle-induced histone gene transcription. We detected much higher levels of

phospho-peptides derived from histone H1.4 in late $G_1$ CK2 complexes than in early $G_1$, where phosphorylated threonine-18 in H1.4 in vivo may be associated with CK2 complexes (Accession No PXD040882). This phosphorylation site was previously reported by high-throughput analysis among upstream kinase(s) contributing to phosphorylation (Bonenfant et al, 2007). Other posttranslational modifications, including citrullination of a single arginine in histone H1, were suggested to contribute to the regulation of pluripotency by displacing H1 from chromatin (Christophorou et al, 2014). The consistently higher phosphorylation of proteins from nuclear fractions than from cytosol (Fig 1C) suggests that phosphorylation is essential for association of nuclear CK2 complexes activating target genes. Our phospho-proteomic analysis revealed numerous novel interactions, thus providing a valuable resource to understand how this kinase signaling pathway affects cellular functions through phosphorylation.

In addition, a number of ribosomal biogenesis components, which are mainly localized in nucleoli, were found in nuclear CK2 complexes, suggesting that CK2 may be involved in rRNA expression in the nucleolus. Association of CK2 with RNA polymerase-I$\alpha$, -I$\beta$ (Lin et al, 2006; Panov et al, 2006; Bierhoff et al, 2008) or -II (Cabrejos et al, 2004) have been reported, and our protein lists showed multiple subunits of Pol I along with those of Pol II (Tables S1 and S2). Mitchell et al demonstrated that CK2 phosphorylated by Akt activated RRN3/TIF-1A, a transcription factor of rDNA, increases rDNA transcription (Nguyen & Mitchell, 2008). rDNA transcription was largely down-regulated in cells depleted of CK2$\alpha$, demonstrating an essential role of CK2 in ribosome biogenesis by association with TIF-1A and multiple sets of DNA topology modifiers, such as RNA helicases, probably helping to tether RNA pol I to the *rDNA* promoter (Fig 6). Moreover, phosphorylation of UBF1 by CK2 is reportedly important, in regard to UBF1 recruitment to *rDNA* loci, and initiation of rRNA transcription (Lin et al, 2006). Because rRNA synthesis is essential for cell proliferation, its aberrant production promotes malignancies. With histochemical analysis of invasive ductal breast cancer specimens, we found that nucleolar localization and accumulation of CK2$\alpha$ was associated with future recurrence (Homma et al, 2021). Results predicting that patients would experience cancer recurrence several years after their primary surgery caught our attention, because CK2 localization in the nucleolus is rarely detected in normal cultured cells or normal tissues using immunohistochemical or biochemical analyses after cell fractionation. Therefore, it is important to examine how CK2$\alpha$ translocates to the nucleolus and whether it contributes to gene expression in that location. When partially purified nuclear fractions from late $G_1$ cells were immunoprecipitated with anti-CK2$\alpha$ antibody, nucleolar proteins that interact with CK2 were identified by mass spectrometry, including RNA-binding proteins, UTP14A, PARP1, Nucleolin, NOLC1, NPM1, UTP18, TCOF1, UTP11, and RNA helicases, DDX9, DDX18, DDX21, DDX39B, DDX17, SMARCA4, DDX5P, DDX52, some of which were verified by Western blotting (Fig 6C). Our results provide evidence for involvement of CK2 in ribosomal gene expression, suggesting that CK2 remodels chromatin to a transcriptionally active state, and confirming a multifunctional role for CK2 in nuclear and nucleolar locations. By exploring molecular connections of DNA topology modifiers, such as RNA helicases and CK2 regarding their involvement in both RNA pol I- and pol II-dependent transcriptional arms of ribosome biogenesis, this study emphasizes the functional significance of CK2 involving a mechanism linking nucleolar function, ribosomal DNA transcription, and cancer progression.

In conclusion, we propose that CK2 participates in cell proliferation through recruitment to the nuclear genome for histone gene activation, and also to the nucleolus for association with ribosomal genes. These CK2 functions appear to be tightly regulated during the cell cycle, with translocation to the nucleus, activation by posttranslational phosphorylation, and CK2–chromatin interactions mediating the transcriptional response. For example, activation of *rDNA* at genomic loci in nucleoli may be associated with cell cycle-dependent localization of CK2$\alpha$. Furthermore, network analysis of this dataset provided a comprehensive overview of protein–protein interactions, probably involving phosphorylation required for gene activation. Therefore, reagents that target *protein interactions* enabling nuclear changes that direct cell growth might be effective for

treatment of cancer. Our data reveal a spatiotemporal role of CK2$\alpha$ as an indispensable molecule for cell proliferation by supporting gene networks and protein synthesis.

# Materials and Methods

### Cell culture

TIG-7 cells are normal human fibroblasts derived from fetal lung established in Tokyo Metropolitan Institute of Gerontology. The cell line demonstrates contact inhibition in culture and finite life span with limited population doublings as normal fibroblast phenotypes (Yamamoto et al, 1991). HEK 293 cells are human embryonic kidney cells. RPE cells are human retinal pigment epithelial cells. TIG-7, HEK293 and RPE cells were grown in DMEM supplemented with 10% FBS. Cells were maintained at 37°C in a humidified 5% $CO_2$ incubator. For synchronization experiments, logarithmically growing cells were starved in 0.1% FBS for 48 h, and then cultured in fresh media containing 10% FBS for an additional 6 or 15–16 h to obtain cell populations enriched in early $G_1$ or late $G_1$, respectively.

### Plasmids and transfections

Full-length cDNAs for human *CK2$\alpha$* subunit were as described previously (Lozeman et al, 1990). Site-directed mutagenesis of CK2 was performed using SiteDirect (Promega) to mutate both Lys 64 and Lys 68 to Ala (KD), and to mutate Ser 7, Ser 194 or Ser 287 to Ala. All constructs and mutations were confirmed by DNA sequencing. For mammalian expression, each cDNA was subcloned into pCDNA3-NFLAG (Invitrogen) and transfected into HEK293 or RPE cells using Effectene reagent (QIAGEN) according to the manufacturer's instructions. For production of recombinant proteins, *CK2$\alpha$* cDNAs were subcloned into pGEX-4T (Amersham Biosciences). Recombinant GST fusion proteins were expressed in *E. coli* strain BL21(DE3), purified by Glutathione–Sepharose chromatography (GE Health Science), and then GST tags were removed with thrombin, according to the manufacturer's protocol. Purity and quantity of recombinant proteins were estimated by SDS/PAGE.

### Kinase assay and in vivo phosphorylation assays

TIG-7 cells were serum-starved to synchronize them in $G_0$, and then stimulated with serum (FBS) for the indicated times, before separating cytosolic and nuclear fractions, followed by immunoprecipitation with anti-CK2$\alpha$ polyclonal antibodies, Lot #6464D, originally from Ed Krebs laboratory at University of Washington, Seattle. CK2 activity was measured using anti-CK2 immunoprecipitates as a kinase source prepared as described, or recombinant proteins after thrombin cleavage, and by p81-filter assay with or without RRREEETEEE as a substrate peptide (Homma et al, 2002). In vivo $^{32}$P-labeling experiments were performed by adding 0.1 mCi of $^{32}$P orthophosphate to each 100-mm culture plate, which was started by splitting $2 \times 10^5$ cells, for the last hour before harvesting cells. Cells were pretreated with

phosphate-free medium during synchronization of the cell cycle. Proteins were separated by SDS–PAGE (7.5% acrylamide) and $^{32}$P incorporation was detected by autoradiography.

## Nuclear extracts, Western blotting, and gel electrophoresis

Synchronously progressing cells were harvested at the indicated times after serum stimulation and fractionated into cytosolic and nuclear fractions, as described (Homma et al, 2015). Briefly, after removal of the cytosolic fraction, nuclear proteins were solubilized with 0.6 M NaCl in 20 mM HEPES (pH 7.4), containing 25 mM $\beta$-glycerophosphate, 20% glycerol, 1 mM DTT, 2 mM EDTA, 1 mg/ml aprotinin, 1 mg/ml leupeptin, and 1 mM PMSF. Nuclear proteins that immunoprecipitated with anti-CK2$\alpha$ or anti-FLAG M2 (#A-1804; Sigma-Aldrich) were added to a kinase reaction mixture to measure kinase activity by P81, or separated by 10% SDS–PAGE, followed by Western blotting.

## Identification of phosphorylation sites in CK2$\alpha$

To identify phosphorylation sites in CK2$\alpha$ in vivo, CK2$\alpha$ proteins expressed in HEK293 cells were immunoprecipitated with anti-CK2 antibodies from nuclear fractions of synchronized early $G_1$ cells (stimulated with serum for 6 h), and concentrated by SDS–PAGE, followed by gel staining and excision of FLAG-CK2$\alpha$ bands, which were de-stained and digested in the gel with 10 ng/$\mu$l trypsin in 20 mM ammonium bicarbonate at 37°C overnight. We performed phosphopeptide enrichment using TiO$_2$ followed by peptide extraction with 50% (vol/vol) acetonitrile containing 0.1% (vol/vol) acetic acid (Homma et al, 2015). LC-MS/MS data acquisition was performed on a 5600 Triple TOF mass spectrometer (ABSciex) interfaced with an Eksigent NanoLC system (Eksigent). For measurements on the 5600 system, peptides were separated on an HiQ Sil C$_{18}$ analytical column (Particle Size: 3 $\mu$m, Pore Size: 120 A: 0.1 mm i.d. × 100 mm, KYA Tech.). The LC system was operated at 300 nl/min with the following buffers: buffer A, 2% acetonitrile, 0.1% formic acid; buffer B, 70% acetonitrile, 0.1% formic acid. The gradient was linear from 2–40% B in 130 min, up to 90% B in 1 min, isocratic at 90% B for 4 min, down to 2% B in 1 min, and isocratic at 2% B for 15 min for mobile phase equilibration, with a total runtime of 150 min. Eluate was delivered into the mass spectrometer with a NanoSpray III source. Gas and other mass spectrometer settings varied depending on optimization. Typical values were curtain gas = 20, Gas 1 = 20, Gas 2 = 0, an ion spray floating voltage around 2,300, and collision energy voltage of 35 V. 15 V of collision energy spread was used for collision induced dissociation (CID) fragmentation for MS/MS spectra acquisition. Each cycle consisted of TOF-MS spectrum acquisition for 250 ms (mass range 400–1,250 kD), followed by acquisition of up to 10 MS/MS spectra (100 ms each, mass range 100–1,600 kD) of MS peaks above intensity 150 taking 1.3 s total per full cycle. Once MS/MS fragment spectra were acquired for a particular mass, that mass was dynamically excluded for 15 s. Acquired data to identify phosphorylation sites in CK2$\alpha$ were processed using ProteinPilot 4.0 software.

## CK2-associated proteins identified by nano-LC mass spectrometry

To identify CK2-associated proteins during cell cycle progression, FLAG-tagged CK2$\alpha$ was expressed in HEK293 cells, and then cytosolic or nuclear fractions were prepared separately from synchronized cells by starvation and restimulation with serum, as described. Each fraction was pre-cleared by extensive incubation with normal IgG–agarose beads and then immunoprecipitation with anti-FLAG M2 agarose beads (#A-2220; Sigma-Aldrich). Proteins adhering to the beads were eluted step-wise with FLAG peptide (10 $\mu$g/ml, #A-3290; Sigma-Aldrich) into five fractions according to the manufacturer's protocol. These fractions were monitored for CK2$\alpha$ protein by Western blotting and fractions containing CK2$\alpha$ were combined, digested with trypsin, and prepared for mass spectrometric analysis as described. Shotgun proteomic analyses were performed with a linear ion trap-orbitrap mass spectrometer (LTQ-Orbitrap Velos; Thermo Fisher Scientific) coupled with a Nanoflow LC System (Dina-2A; KYA Technologies). Peptides were injected into a 75-$\mu$m reversed-phase C$_{18}$ column at a flow rate of 10 $\mu$l/min and eluted with a linear gradient of solvent A (2% acetonitrile and 0.1% formic acid in H$_2$O) to solvent B (40% acetonitrile and 0.1% formic acid in H$_2$O) at 300 nl/min. Peptides were sequentially sprayed from a nanoelectrospray ion source (KYA Technologies) and analyzed by CID. Analyses were operated in data-dependent mode, switching automatically between MS and MS/MS acquisition. For CID analyses, full-scan MS spectra (from m/z 380 to 2,000) were acquired in the Orbitrap with a resolution of 100,000 at m/z 400 after ion count accumulation to a target value of 1,000,000. The 20 most intense ions at a threshold above 2,000 were fragmented in the linear ion trap with normalized collision energy of 35% for activation time of 10 ms. The orbitrap analyzer was operated with the "lock mass" option to perform shotgun detection with high accuracy (Thermo Fisher Scientific). RAW files were converted into complete peak lists using the Xcalibur software package, and for each MS/MS spectrum, fragment ions were converted into .mgf files without further data processing, and were then entered into Proteome Discoverer (ver.1.3; Thermo Fisher Scientific) for database searching. Protein identification was performed by automated database searches using the Mascot algorithm (ver. 2.4; Matrix Science) and the SWISS–PROT database, release 57.15 with the following parameters. Enzyme specificity was fully tryptic, allowing up to two missed cleavages. N-acetylation, methionine oxidation, pyroglutamination (Gln), and phosphorylation (Ser, Thr and Tyr) were included as variable modifications. Protein identification was based on peptide thresholds with a 99.0% confidence minimum; peptide cutoff scores greater than 10, PSMs with deltaCn scores better than 10, protein score thresholds (Mascot) greater than 29, and protein significance with a 99.0% confidence minimum. An additional criterion (scores >29) to evaluate identification was applied for single-peptide identification. EmPAI calculation was performed to obtain semiquantitative estimates of protein changes based on spectral count data from Mascot (Shinoda et al, 2010). In order to compare protein levels between groups, cell culture plates in early $G_1$ and late $G_1$ had been inoculated with the same number of cells, and we normalized the

two sets according to total spectral counts. A FDR < 0.05 was used. Raincloud plots were generated using R (Allen et al, 2021).

## ChIP

RPE cells ($1–2 \times 10^7$ cells) synchronously growing through $G_1$ phase were fixed with 1% formaldehyde for 10 min for PolII, H3K4me3, H3K27ac, or PolIISer2P ChIP, and with di-succinimidyly-glutarate for 45 min, followed by 10 min fixation with 1% formaldehyde for CK2$\alpha$ ChIP. Fixation was performed at RT and terminated by incubation with 125 mM glycine for 10 min. Fixed chromatin was sonicated in lysis buffer containing 20 mM Tris–HCl, pH 7.5, 150 mM NaCl, 1% Triton X-100, 1% NP-40, 1% sodium cholate, 0.1% SDS, using a Branson Sonifier 250D to generate DNA fragments with a mean length of 300–1,000 bp, clarified by centrifugation and pre-cleared with protein A agarose beads. Resulting supernatants were then immunoprecipitated with 2 $\mu$g of antibody, monoclonal anti-CK2 (#70774; Abcam) for CK2-ChIP, for each sample by incubating for 15 h at 4°C. The chromatin-antibody mix was further incubated with 50 $\mu$l of Protein A- or Protein G-agarose beads for 4 h at 4°C. Beads were washed vigorously with increasing concentrations of NaCl (150, 300 and 500 mM) and material captured on the beads was eluted with Tris–EDTA containing 1% SDS. Crosslinking was reversed overnight at 65°C, and DNA was purified using QIAGEN columns. All buffers included 1mM PMSF, 1 $\mu$g/$\mu$l aprotinin, 1 $\mu$g/$\mu$l leupeptin, 2 mM Na$_3$VO$_4$, 10 mM $\beta$-glycerol phosphate, and 10 mM NaF.

## ChIP-seq analysis

High-throughput sequencing was performed using a HiSeq 2500 system (Illumina). Input and ChIP DNA was processed and sequenced according to the manufacturer's instructions for NEBNext Ultra II DNA Library Prep kit for Illumina (New England Biolabs). Briefly, DNA was sheared to an average size of ~150 bp with ultrasonication (Covaris), end-repaired, ligated to sequencing adapters, amplified, size-selected, and DNA was then sequenced to generate single-end, 65-bp reads using the Illumina HiSeq-2500 systems. Reads were aligned to the human (UCSC hg38) genome using Bowtie 2 (Li & Dewey 2011; Langmead & Salzberg, 2012) and mapped reads were processed with DROMPAplus (Nakato & Sakata, 2021). In peak calling, both ChIP and input sequence results were used to eliminate pseudo-binding signals in input samples. ChIP-sequencing data for CK2, H3K4m3, H3K27ac, Pol II, and Pol II Ser-P2, were deposited in Gene Expression Omnibus database under entry GSE226778.

For motif finding and enrichment analysis, adapter trimming was performed using Trimmomatic-0.39, and read quality was assessed with FastQC. Reads were aligned to the hg38 reference genome using Bowtie2 with default parameters (Carlson & Maintainer, 2015). Macs2 peak caller version 2.1.2 was employed to identify peaks (Zhang et al, 2008). The minimum $Q$-value was set to 0.01. The broad cutoff was set to 0.1. To determine thresholds for significant peaks, data were manually inspected in IGV 2.6.2. After conversion of BAM files to Bigwig format with deepTools bamCoverage, log-transformed counts per sample were computed with deepTools multiBamsammary (Fidel et al, 2016). Pearson correlation was employed to reveal the degree of similarity between samples using

deepTools plotCorrelation. The "complete" method was adopted for hierarchical clustering. For mapping of CK2–ChIP-seq reads (our data) along with DDX21–ChIP-seq reads (Calo et al, 2015) to the human $rDNA$, the DNA consensus sequence of the unique 43-kb ribosomal locus NCBI (GeneBank: U13369.1) was obtained, and the Bowtie algorithm was used. Parameters were set as default and hg38 was used as the reference genome sequence. The analysis procedure combining Chr 13 centromere and ribosomal DNA repeats has been the same as employed by previous studies (Zentner et al, 2011; Cong et al, 2012).

## Global gene expression analysis

Comprehensive gene expression analysis was performed according to previous reports (Miura et al, 2006; Higa et al, 2017). Briefly, synthetic polynucleotides (80-mers) representing 14,400 human transcripts (MicroDiagnostic) were arrayed using a custom arrayer. Total RNA was extracted from cells using ISOGEN (Nippon Gene) and 5 $\mu$g were labeled using SuperScript II (Invitrogen) and cyanine 5-dUTP (PerkinElmer) for samples or cyanine 3-dUTP (PerkinElmer) for human common reference RNA, which was prepared by mixing equal amounts of total RNA extracted from 22 cell lines. Hybridization was performed with a labeling and hybridization kit (MicroDiagnostic). Signals were measured with a GenePix 4000B scanner (Molecular Devices). Signals were converted into primary expression ratios (ratio of cyanine-5 intensity of each sample to cyanine-3 intensity of the human common reference RNA). Each ratio was normalized through multiplication using normalization factors and Gene Pix Pro 3.0 software (Molecular Devices). The primary expression ratio was converted to log$_2$ values (designated as log ratios). Spots that exhibited fluorescence intensities below the detection limit were assigned a log ratio value of 0 and were not included in signal calculations of averages and subtracted log ratios. Data were processed using Microsoft Excel software (Microsoft) and the MDI gene expression analysis software package (MicroDiagnostic). Raw data files have been deposited in NCBI GEO with the dataset number, GSE227739.

## Quantitative PCR

ChIP-purified DNA was analyzed by qPCR using a real-time PCR system StepOne (Applied Biosystems) and KAPA SYBR Fast qPCR kit (KAPA Biosystems). Primer sets used are listed.

## CRISPR-Cas9 was used for establishment of CK2$\alpha$ knock-out cell clones

For genome editing mediated by CRISPR-Cas9, sgRNAs were designed using the CRISPR design program (Ran et al, 2015). sgRNA guide sequences were 5′-TAGTCACGAACCCCCGCCTT-3′ or 5′- CGTAAACAA-CACAGACTTCA-3′ and were subcloned into pSpCas9n(BB) vectors (PX462 v2.0; Addgene). RPE cells were cultured to 70% confluence and transfected with 1 $\mu$g of sgRNA plasmids (0.5 $\mu$g each) with the Neon Transfection System, according to the manufacturer's instructions (Invitrogen). After 72 h, limited-diluted cells were split into 96-well plates at a concentration of 0.5 cells per well. Single clones of CK2$\alpha$-ko cells were isolated and the absence of CK2$\alpha$ was validated by

Western blotting and direct DNA sequencing (BigDyeTerminator, v3.1; Thermo Fisher Scientific) on PCR products amplified from genomic DNA spanning the sgRNA target site, along with mass spectrometry analysis using TMT-labeling reagent (Thermo Fisher Scientific). The following primers pairs were used: forward 5′ -CAGCATCCGCAAA-CATTG-3′ and reverse 5′-CTCTTCTAACAGCATCATCCCC-3′ primers to amplify the ko-CK2α sequence.

## Statistical analysis

Data in two groups were evaluated with unpaired two-tailed $t$ tests, and multiple comparison tests were performed with one-way analysis of variance followed by Turkey's multiple comparison test. Values representing means ± SDs are represented by error bars. Statistical significance is as indicated. A $P$-value < .05 was considered significant. All statistical analyses and structuring were performed using JMP Pro v.14.2.0 (SAS), except the Pearson's correlation coefficient or Wilcoxon signed-rank test, as noted in figure legends.

# Data Availability

Mass spectrometry proteomics data have been deposited to the ProteomeXchange Consortium (http://proteomecentral.proteomexchange.org) via the PRIDE partner repository (Vizcaino et al, 2013) and assigned the identifier PXD040882, and also via the jPOST repository (Okuda et al, 2017) with the dataset identifier JPST002085. CK2–ChIP-seq data have been deposited in NCBI GEO with the dataset number GSE226778. Global gene expression array data have been deposited in NCBI GEO with the dataset number GSE227739.

# Supplementary Information

# Acknowledgements

We are indebted to Dr. Noriko Saitoh for fruitful discussions. We thank Dr. Shio Watanabe for assistance with LC-MS instrumentation, and Dr. Satoru Moritoh for introduction of the CRISPR-Cas9 system. We thank the Biostatistical Consulting Service at the Clinical Research Center, Fukushima Medical University, and in particular, Dr. Noriko Tanaka, who helped with interpretation of results of statistical analyses in this study. MK Homma gratefully acknowledges funding by Japan Agency for Medical Research and Development (AMED) under Grant numbers 20lm0203006j0004 and 22ym00126808j0001, and support by Basis for Supporting Innovative Drug Discovery and Life Science Research from AMED. This work was supported by grants-in-aid from the Ministry of Education, Culture, Sports, Science and Technology of Japan, by the Grant for Joint Research Project of the Institute of Medical Science, the University of Tokyo, and by the Research Seeds Quest Program of the Japan Science and Technology Agency.

## Author Contributions

MK Homma: conceptualization, data curation, formal analysis, funding acquisition, validation, investigation, visualization, methodology, project administration, and writing—original draft, review, and editing.
R Nakato: data curation, formal analysis, validation, visualization, and methodology.
A Niida: data curation, validation, visualization, methodology, and writing—original draft.
M Bando: data curation, formal analysis, investigation, and methodology.
K Fujiki: data curation and formal analysis.
N Yokota: data curation and formal analysis.
S Yamamoto: data curation, formal analysis, visualization, and writing—original draft.
T Shibata: data curation, formal analysis, and methodology.
M Takagi: data curation, formal analysis, and investigation.
J Yamaki: data curation, formal analysis, investigation, and methodology.
H Kozuka-Hata: data curation, formal analysis, validation, investigation, and methodology.
M Oyama: data curation, formal analysis, validation, investigation, and methodology.
K Shirahige: conceptualization, resources, data curation, supervision, and project administration.
Y Homma: conceptualization, resources, supervision, validation, and writing—original draft and project administration.

## Conflict of Interest Statement

The authors declare that they have no conflict of interest.

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
