## [Reviewer comments · Life Science Alliance]

Life Science Alliance

Cell cycle-dependent gene networks for cell proliferation activated by nuclear CK2 alpha complexes

Miwako Homma, Ryuichiro Nakato, Atsushi Niida, Masashige Bando, Katsunori Fujiki, Naoko Yokota, So Yamamoto, Takeshi Shibata, Motoki Takagi, Junko Yamaki, Hiroko Kozuka-Hata, Masaaki Oyama, Katsuhiko Shirahige, and Yoshimi Homma
DOI: <https://doi.org/10.26508/lsa.202302077>

Corresponding author(s): Miwako Homma, Fukushima Medical University

Review Timeline:

Submission Date:	2023-04-05
Editorial Decision:	2023-05-30
Revision Received:	2023-09-02
Editorial Decision:	2023-10-02
Revision Received:	2023-10-09
Accepted:	2023-10-10

Transaction Report:

May 30, 2023

Re: Life Science Alliance manuscript #LSA-2023-02077-T

Dr Miwako Kato Homma
Fukushima Medical University School of Medicine
Bimolecular Sciences
1-Hikari-gaoka
Fukushima 9601295
Japan

Dear Dr. Homma,

Thank you for submitting your manuscript entitled "Cell cycle-dependent gene networks for cell proliferation activated by nuclear CK2 complexes" to Life Science Alliance. The manuscript was assessed by expert reviewers, whose comments are appended to this letter. We invite you to submit a revised manuscript addressing the Reviewer comments.

Thank you for this interesting contribution to Life Science Alliance. We are looking forward to receiving your revised manuscript.

Sincerely,

B. MANUSCRIPT ORGANIZATION AND FORMATTING:

Reviewer #1 (Comments to the Authors (Required)):

General Comments:

In this manuscript, the authors conduct a detailed examination of the activation and function of nuclear CK2 during cell proliferation. Multiple phosphorylation sites were examined, and the association of CK2alpha with transcriptional start sites was studied in detail. The work involves both proteomic and ChIP-seq approaches. The paper is very clearly written. It represents a major piece of work that advances the field in multiple aspects.

2. The data are strongly supportive of all of the conclusions that are drawn, with the caveat of the issue raised below concerning reproducibility.

3. There are several issues, most involving data presentation:

- a. The abstract requires English editing. In addition, an abstract typically includes more information. For example, in the sentence beginning "This phosphorylation appears..." there is no indication of what types of experiments led to the conclusion; also the conclusion is rather weakly stated.
- b. There needs to be more attention to scientific reproducibility. The reviewer did not see any use of statistics in the paper. This reviewer is not a believer in excessive use of statistics, but many of the figures look odd without it (e.g., Figure 1F) as the addition of statistical information helps to direct the reader to the most important parts of each figure. The authors do not indicate how many replicate experiments were performed, or how many replicate measurements are reflected in each error bar. Overall, this is a very important issue that must be addressed.
- c. All of the labels that read "Blot with CK2alpha" should be "Blot for..." or better yet just "CK2alpha". They need to be consistent...Figure 3A is labelled "anti-CK2alpha".
- d. In Figure 6C, it is not apparent from the images what is being blotted for.

Reviewer #2 (Comments to the Authors (Required)):

This manuscript describes the control of nuclear translocation of the protein kinase, CK2alpha by phosphorylation. Evidence is presented showing that (i) CK2a is phosphorylated at Ser7, S194 and S287; (ii) pSer7 appears necessary for nuclear translocation and serum stimulated cell proliferation; (iii) CK2 ChIP seq identifies promoter and genic loci for genes controlled in late G1 and ribosome biogenesis; (v) CK2a coimmunoprecipitates with regulators of late G1 and rRNA transcription, including RNAPII subunits, histones, and RNA helicases. The findings are taken to suggest that CK2 may facilitate G1 progression and ribosome biogenesis by associating with nuclear and nucleolar transcriptional regulators. Therefore, the study supports and expands new knowledge by identifying candidate targets for nuclear CK2a.

Strengths of the study are the numerous screens comparing WT, CK2-knock out, and CK2-S7A cells by transcriptome profiling, ChIP-seq, and co-immunoprecipitation. Experiments are described adequately and the data analysis appears sound. This is a rich dataset that will be useful to many researchers for understanding nuclear function(s) of CK2, which so far have been elusive. That many of these screens converge on genes involved in late G1 and rRNA transcription will be of particular interest to this field. Weaknesses are that the findings are mostly correlative. Conceivably, some of the transcriptional or co-IP responses that are altered in CK2-KO are secondary effects of the KO on cell cycle inhibition. Nevertheless, the data resource alone represent an impressive amount of work, and the co-IP data nicely ties the screening results together to support the hypothesis that CK2 functions through direct interactions with RNAPII and transcription regulators. Overall, the data generated by this paper should be valuable for understanding CK2 mechanisms and explaining the yet poorly understood requirement for this kinase in cell cycle progression.

Reviewer #1 (Comments to the Authors (Required)):

General Comments:

In this manuscript, the authors conduct a detailed examination of the activation and function of nuclear CK2 during cell proliferation. Multiple phosphorylation sites were examined, and the association of CK2alpha with transcriptional start sites was studied in detail. The work involves both proteomic and ChIP-seq approaches. The paper is very clearly written. It represents a major piece of work that advances the field in multiple aspects.

2. The data are strongly supportive of all of the conclusions that are drawn, with the caveat of the issue raised below concerning reproducibility.

We thank the Reviewer for his/her positive feedback on our manuscript. We have responded to all comments below.

3. There are several issues, most involving data presentation:

a. The abstract requires English editing. In addition, an abstract typically includes more information. For example, in the sentence beginning "This phosphorylation appears..." there is no indication of what types of experiments led to the conclusion; also the conclusion is rather weakly stated.

We thank the reviewer for calling this to our attention. To address this comment, we have revised the Abstract. It now reads as follows.

Nuclear expression of protein kinase CK2 α is reportedly elevated in human carcinomas, but mechanisms underlying its variable localization in cells are poorly understood. This study demonstrates a functional connection between nuclear CK2 and gene expression in relation to cell proliferation. Growth stimulation of quiescent fibroblasts and phospho-proteomic analysis identified a pool of CK2 α that is highly phosphorylated at serine 7. Phosphorylated CK2 α translocates into the nucleus, and this phosphorylation appears essential for nuclear localization and catalytic activity. Protein signatures associated with nuclear CK2 complexes reveal enrichment of apparently unique transcription factors and chromatin remodelers during progression through G₁ phase of the cell cycle. Chromatin immunoprecipitation-sequencing profiling demonstrated recruitment of CK2 α to the active gene loci, more abundantly in late G₁ phase than in early G₁, notably at transcriptional start sites of core histone genes, growth stimulus-associated genes, and ribosomal RNAs. Our findings reveal that nuclear CK2 α complexes may be essential to facilitate progression of the cell cycle, by activating histone genes,

triggering ribosome biogenesis, specified in association with nuclear and nucleolar transcriptional regulators.

b. There needs to be more attention to scientific reproducibility. The reviewer did not see any use of statistics in the paper. This reviewer is not a believer in excessive use of statistics, but many of the figures look odd without it (e.g., Figure 1F) as the addition of statistical information helps to direct the reader to the most important parts of each figure. The authors do not indicate how many replicate experiments were performed, or how many replicate measurements are reflected in each error bar. Overall, this is a very important issue that must be addressed.

We agree. To address this comment, we checked the original data and added statistical information to the figures. We also added numbers of replicate experiments and replicate measurements to the figure legends. Additionally, Figure EV2A has now been corrected, after re-calculating our ChIP-Seq data. This re-calculation did not affect our interpretation of the results. Happily, it strengthened correlations between the two factors. It can be easily reproduced by the reader, using deepTools algorithms in standardization methods and procedures.

c. All of the labels that read "Blot with CK2alpha" should be "Blot for..." or better yet just "CK2alpha". They need to be consistent...Figure 3A is labelled "anti-CK2alpha".

We thank the Reviewer for pointing this out. We have made the suggested alterations by changing the label "Blot with CK2 α " to just "CK2 α ." Concerning Figure 3D, which demonstrates association of RNA polymerase II with CK2a, a blot for CK2a is now included in order to better support the conclusions.

d. In Figure 6C, it is not apparent from the images what is being blotted for.

We acknowledge that Figure 6C was poorly labeled. We have added the name of the target molecule ("Flag") to the blot in the figure.

Reviewer #2 (Comments to the Authors (Required)):

This manuscript describes the control of nuclear translocation of the protein kinase, CK2alpha by phosphorylation. Evidence is presented showing that (i) CK2a is phosphorylated at Ser7, S194 and S287; (ii) pSer7 appears necessary for nuclear translocation and serum stimulated cell proliferation; (iii) CK2 ChIP seq identifies promoter and genic loci for genes controlled in late G1 and ribosome biogenesis; (v) CK2a coimmunoprecipitates with regulators of late G1 and rRNA transcription, including RNAPII subunits, histones, and RNA helicases. The findings are taken to suggest that CK2 may facilitate G1 progression and ribosome biogenesis by associating with nuclear and nucleolar transcriptional regulators. Therefore, the study supports and expands new knowledge by identifying candidate targets for nuclear CK2a.

Strengths of the study are the numerous screens comparing WT, CK2-knock out, and CK2-S7A cells by transcriptome profiling, ChIP-seq, and co-immunoprecipitation. Experiments are described adequately and the data analysis appears sound. This is a rich dataset that will be useful to many researchers for understanding nuclear function(s) of CK2, which so far have been elusive. That many of these screens converge on genes involved in late G1 and rRNA transcription will be of particular interest to this field. Weaknesses are that the findings are mostly correlative. Conceivably, some of the transcriptional or co-IP responses that are altered in CK2-KO are secondary effects of the KO on cell cycle inhibition. Nevertheless, the data resource alone represent an impressive amount of work, and the co-IP data nicely ties the screening results together to support the hypothesis that CK2 functions through direct interactions with RNAPII and transcription regulators. Overall, the data generated by this paper should be valuable for understanding CK2 mechanisms and explaining the yet poorly understood requirement for this kinase in cell cycle progression.

We thank the Reviewer for this positive comment on our manuscript. Concerning the comment, "Weaknesses are that the findings are mostly correlative. Conceivably, some of the transcriptional or co-IP responses that are altered in CK2-KO are secondary effects of the KO on cell cycle inhibition", we agree. We found that CK2 concentrates in proximity to promoters as the cell cycle progresses and actively participates in gene expression by interacting with transcription factors and Pol2. However, how CK2 cooperates in gene expression with RNA Polymerase II is uncertain. We do not know whether CK2 directly binds to DNA like a transcription factor, activates RNA Polymerase II by phosphorylation, activates RNA Polymerase II by phosphorylating and removing regulatory factors, or phosphorylates histone proteins or chromatin remodelers to modify higher-order structure of chromatin, etc. Therefore, it will be important to clarify details of the

molecular function of CK2 in proximity to promoters. We hope to publish the results as we proceed with verification. Fortunately, we have established unique, high-quality, CK2-specific antibodies, clones 6A and 16C, and analysis using them is expected to yield valuable insights.

October 2, 2023

RE: Life Science Alliance Manuscript #LSA-2023-02077-TR

Dr. Miwako Kato Homma
Fukushima Medical University
Bimolecular Sciences
1-Hikari-gaoka
Fukushima, Fukushima 9601295
Japan

Dear Dr. Homma,

Thank you for submitting your revised manuscript entitled "Cell cycle-dependent gene networks for cell proliferation activated by nuclear CK2 complexes". We would be happy to publish your paper in Life Science Alliance pending final revisions necessary to meet our formatting guidelines.

- please address Reviewer 1's remaining minor points
- please upload all figure files as individual ones, including the supplementary figure files
- LSA allows supplementary figures/tables, but no EV Figures/Tables; please update your callouts for the Supplementary Figures/Tables in the manuscript Fig EV1A=Fig S1A, Table EV1=Table S1; while supplementary figures/tables use the system supplementary Fig S1/Table S1
- please add the Twitter handle of your host institute/organization as well as your own or/and one of the authors in our system
- please note that titles in the system and on the manuscript file must match
- There is a name discrepancy in the name presentation of your co-author. Please correct: in the ms file -- Atsushi Niita vs. system Niida Atsushi
- please use the [10 author names et al.] format in your references (i.e., limit the author names to the first 10)
- please upload one file per figure

Figure checks:

- please indicate sizes next to each blot
- please add scale bars to Figures 1D and S3C

A. FINAL FILES:

B. MANUSCRIPT ORGANIZATION AND FORMATTING:

Sincerely,

Reviewer #1 (Comments to the Authors (Required)):

General:

The authors have been very responsive to the points raised in the earlier review. The reproducibility aspect of the manuscript has been much improved in the process. A few minor edits are mentioned below; these edits can be made at the proofing stage.

Minor Points:

1. Line 41: The species/source of the fibroblasts should be stated.
2. Line 43: should be "translocates"
3. Line 49: should be "...genes and triggering ribosomal biogenesis, ..."
4. Line 65: Should be "A phosphoproteomic study..."
5. Line 85: Should be "...normal fibroblast..."

Reviewer #1 (Comments to the Authors (Required)):

General:

The authors have been very responsive to the points raised in the earlier review. The reproducibility aspect of the manuscript has been much improved in the process. A few minor edits are mentioned below; these edits can be made at the proofing stage.

Minor Points:

1. Line 41: The species/source of the fibroblasts should be stated.
2. Line 43: should be "translocates"
3. Line 49: should be "...genes and triggering ribosomal biogenesis, ..."
4. Line 65: Should be "A phosphoproteomic study..."
5. Line 85: Should be "...normal fibroblast..."

We thank the Reviewer for his/her positive feedback on our manuscript, and for constructive suggestions. We have responded to all comments below.

1. Human normal fibroblasts, TIG-7, which we have used for cell cycle-dependent CK2 translocation to the nucleus, demonstrates contact inhibition in culture and a finite life span with a limited number of mitotic divisions as a normal fibroblast phenotype. This line was established in 1975 at Tokyo Metropolitan Institute of Gerontology. We have corrected the Abstract accordingly, and included these explanation in the Materials and Methods. We have also added a paper concerning these cells to the Reference section:
Yamamoto K, Kaji K, Kondo H, Matsuo M, Shibata Y, Tasaki Y, Utakoji T, Ooka H. A new human male diploid cell strain, TIG-7 (1991) *Experimental Gerontology* **26**: 525-540.

2. to 5.
We have made the suggested alterations by changing the corresponding.

Once again, we thank the Reviewer for these useful suggestions.

October 10, 2023

RE: Life Science Alliance Manuscript #LSA-2023-02077-TRR

Dr. Miwako Kato Homma
Fukushima Medical University
Bimolecular Sciences
1-Hikari-gaoka
Fukushima, Fukushima 9601295
Japan

Dear Dr. Homma,

Thank you for submitting your Research Article entitled "Cell cycle-dependent gene networks for cell proliferation activated by nuclear CK2 alpha complexes". It is a pleasure to let you know that your manuscript is now accepted for publication in Life Science Alliance. Congratulations on this interesting work.

DISTRIBUTION OF MATERIALS:

Again, congratulations on a very nice paper. I hope you found the review process to be constructive and are pleased with how the manuscript was handled editorially. We look forward to future exciting submissions from your lab.

Sincerely,
